# LInK: Learning Joint Representations of Design and Performance Spaces through Contrastive Learning for Mechanism Synthesis

**Amin Heyrani Nobari**                                                *ahnobari@mit.edu*
*Department of Mechanical Engineering*
*Massachusetts Institute of Technology*

**Akash Srivastava**                                          *akash.srivastava@ibm.com*
*MIT-IBM Watson AI Lab*
*IBM Research*
*Cambridge, Massachusetts 02142*

**Dan Gutfreund**                                                  *dgutfre@us.ibm.com*
*MIT-IBM Watson AI Lab*
*IBM Research*
*Cambridge, Massachusetts 02142*

**Kai Xu**                                                               *xuk@ibm.com*
*MIT-IBM Watson AI Lab*
*IBM Research*
*Cambridge, Massachusetts 02142*

**Faez Ahmed**                                                          *faez@mit.edu*
*Department of Mechanical Engineering*
*Massachusetts Institute of Technology*

**Reviewed on OpenReview:** *https://openreview.net/forum?id=a1MRjOL6WJ*

## Abstract

In this paper, we introduce LInK, a novel framework that integrates contrastive learning of performance and design space with optimization techniques for solving complex inverse problems in engineering design with discrete and continuous variables. We focus on the path synthesis problem for planar linkage mechanisms. By leveraging a multimodal and transformation-invariant contrastive learning framework, LInK learns a joint representation that captures complex physics and design representations of mechanisms, enabling rapid retrieval from a vast dataset of over 10 million mechanisms. This approach improves precision through the warm start of a hierarchical unconstrained nonlinear optimization algorithm, combining the robustness of traditional optimization with the speed and adaptability of modern deep learning methods. Our results on an existing benchmark demonstrate that LInK outperforms existing methods with 28 times less error compared to a state-of-the-art approach while taking 20 times less time on an existing benchmark. Moreover, we introduce a significantly more challenging benchmark, named LINK-ABC, which involves synthesizing linkages that trace the trajectories of English capital alphabets—an inverse design benchmark task that existing methods struggle with due to large nonlinearities and tiny feasible space. Our results demonstrate that LInK not only advances the field of mechanism design but also broadens the applicability of contrastive learning and

optimization to other areas of engineering. The code and data are publicly available at
`https://github.com/ahnobari/LInK`.

# 1 Introduction

Linkage mechanisms play a pivotal role in mechanical engineering, serving as fundamental components in applications ranging from the automation of manufacturing processes to the development of robotic systems. These mechanisms are crucial for translating input motions—often rotational—into desired output trajectories or paths, thereby enabling the execution of complex tasks through relatively simple inputs. The design of linkage mechanisms relies heavily on kinematics the study of motion without considering the forces involved and kinematic synthesis, the design process of mechanisms to achieve specific motions.

Despite the reliance on the established practices of kinematics and kinematic synthesis, designing complex kinematic systems remains a challenge Lipson (2008). The design process often requires trial and error, specialized expertise, or heuristic approaches to find effective solutions. The complexity arises because kinematic synthesis requires defining both the discrete components of a mechanism and their connections, as well as determining their continuous spatial locations, making it a mixed combinatorial and continuous problem.

This paper focuses on the inverse design of planar linkage mechanisms, where the design involves determining how many joints a mechanism has, what type of joints they are (either fixed or revolute joints, and how these joints are connected to one another using rigid linkages. Further, the initial position of the joints in space must be determined. At each of these levels, mechanisms can be infeasible, and the majority of randomly sampled mechanisms tend to be invalid Heyrani Nobari et al. (2022). By examining the continuous values of initial positions, the nonlinear and discontinuous nature of the problem becomes apparent. For instance, Figure 3 illustrates a mechanism that writes the letter B, where all design variables are held constant except for one joint (highlighted in red), which is moved in the x and y directions to demonstrate the feasible regions for these variables. The red-shaded area indicates where the mechanism fails to function. The three grey patches of feasibility for just one joint demonstrate the problem's nonlinearity and discontinuity, featuring multiple feasible regions. Furthermore, adjusting one joint alters the shape and number of feasible regions for all others. This discussion has focused solely on the continuous variables of joint positioning, yet the combinatorial challenge of defining the mechanism's structure also plays a critical role. Not surprisingly, kinematic synthesis is sometimes regarded more as an art than a science due to its complexities 10. (2018).

In this paper, we introduce a contrastive learning and optimization framework to address such inverse design challenges, exemplifying broader issues in kinematic synthesis and engineering design that involve integrating continuous and discrete variables within functional and engineering constraints. Our approach not only addresses existing difficulties in kinematic design but also leverages deep learning to drive advancements in mechanical systems (Regenwetter et al., 2022). A significant challenge in applying deep learning to engineering is the accurate capture of physics-based performance, achieving precision in meeting requirements, effective representation of design and performance spaces, and the mitigation of data scarcity.

To demonstrate the effectiveness of our proposed framework, we focus on a specific type of kinematic synthesis, called, the path synthesis problem. This problem involves designing the aforementioned linkage mechanisms such that they can trace a desired curve with one circular motion of a single actuator. Planar linkage mechanisms use revolute and prismatic pairs to produce rotating, oscillating, or reciprocating motions which can combine into complex motions that trace desired curves. These mechanisms are commonly found in everyday machinery, such as in the moving parts of printing presses, auto engines, and robotics components, making their design critical to a wide range of applications.

Most research on the inverse design of planar linkage mechanisms has applied deep learning to a limited problem of only generating a specific set of mechanism types (typically, four-bar and six-bar mechanisms) (Cabrera et al., 2002; Varedi-Koulaei & Rezagholizadeh, 2020; Ebrahimi & Payvandy, 2015; 10., 2010; Khan et al., 2015b; McGarva, 1994; Chu & Sun, 2010a; Deshpande & Purwar, 2019a; Vasiliu & Yannou, 2001; Deshpande & Purwar, 2019b; 2020; Khan et al., 2015a). In other cases, deep learning frameworks such as deep reinforcement learning, are applied to one task (a single target curve) at a time, only to find a single solution

for a given desired curve (Fogelson et al., 2023) instead of quickly generating a set of solutions for multiple target curves. Additionally, as we show later, existing methods lack high precision and have large errors in achieving the target.

To tackle these challenges of path synthesis in planar linkage mechanisms, we introduce Learning-accelerated Inverse Kinematics (LInK), which synergizes optimization with deep learning—termed "deep optimization"– to accelerate the process and surpass traditional methods in performance. Our approach utilizes a multi-modal, transformation-invariant contrastive learning framework to accurately represent and integrate the different modalities of performance and design spaces, enabling rapid retrieval of mechanisms from extensive datasets. Specifically, we train a multimodal contrastive learning model on a dataset of 10 million linkage mechanisms and their simulated kinematics, precompute embeddings for these mechanisms and later retrieve them using embeddings of target curves produced by the same model (see Figure 4). Additionally, we refine these mechanisms through a hierarchical optimization algorithm, leveraging optimization's inherent robustness and accuracy to improve the precision of our generated mechanisms. This integration of advanced engineering design techniques allows us to excel in designing linkage mechanisms, significantly surpassing existing methods in both efficiency and precision with 28 times better performance and a 20 times increase in speed compared to prior works on an established benchmark. We also create a significantly more challenging benchmark and show that our method can find additional solutions within this new benchmark.

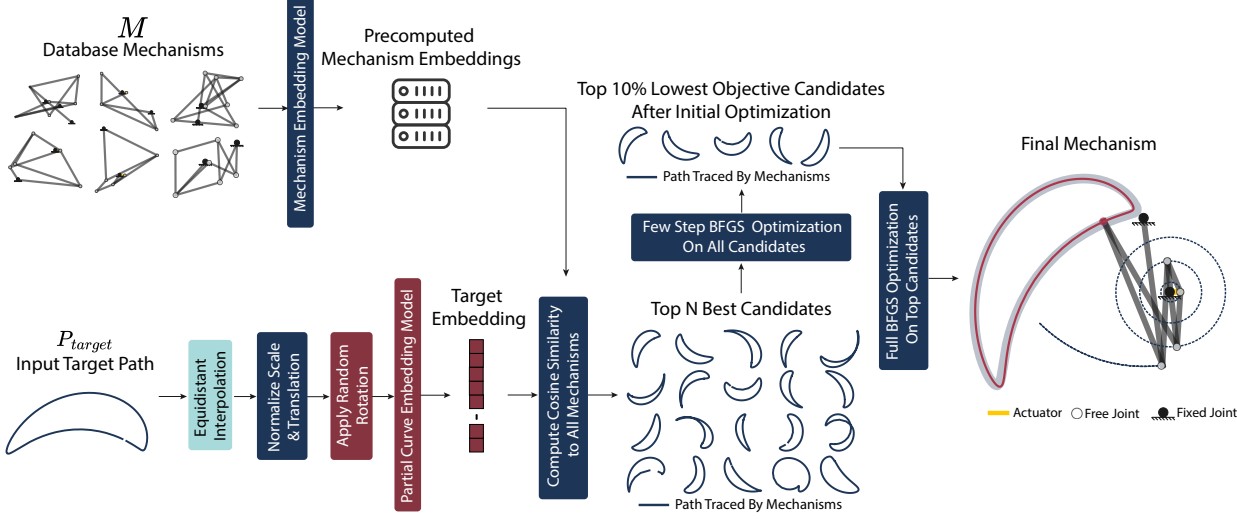

Figure 1: This figure illustrates the LInK algorithm's workflow. LInK first precomputes joint embeddings for mechanisms and curves, allowing it to retrieve numerous candidate mechanisms for any new target curve within seconds. With these high-quality initial candidates, a rapid BFGS optimization process efficiently converges to path synthesis solutions. These solutions significantly outperform existing methods in both speed and performance.

Our key contributions in this paper can be summarized as follows:

- We introduce a contrastive learning approach to learn joint embeddings of design and performance space for problems with both continuous and discrete parameters.

- We introduce a hierarchical deep optimization framework for integrating multimodal contrastive learning-based warm-start with traditional optimization for inverse kinematic design of planar linkage mechanisms. We show that the proposed approach significantly outperforms the state-of-the-art in both speed and performance.

- We develop a graph neural network framework named 'Hop Attention,' designed to effectively capture features sensitive to the number of hops within a graph neural network. This feature is essential for analyzing planar linkage mechanisms.

- We propose a new set of benchmark problems for inverse kinematics in planar linkage mechanisms, advancing beyond existing benchmarks with two new problems that introduce significantly higher levels of complexity and exceed the simple shapes produced by less complex mechanisms.

## 2  Background and Related Works

This section outlines relevant literature and contextualizes the research presented here. We begin with an overview of the specific problem we aim to address and its inherent complexities. We then review traditional computational methods and recent learning-based advancements applied to this problem. Finally, we discuss prior works that have influenced our framework and the methodologies it incorporates.

### 2.1  Inverse Kinematics and The Path Synthesis Problem

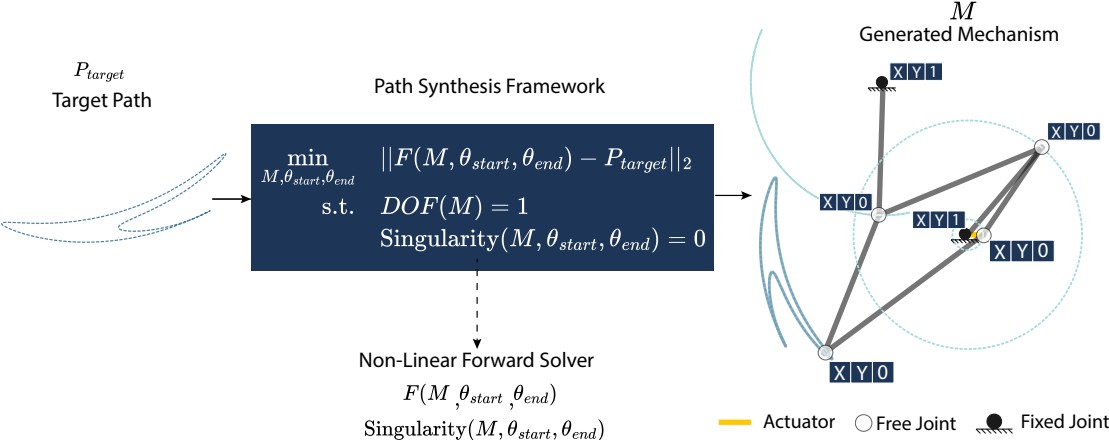

Figure 2: This figure demonstrates the path synthesis problem which involves mixed combinatorial and continuous values. This figure also shows how mechanisms are represented as graphs.

**Inverse Kinematics:** Inverse kinematics is a computational technique primarily used in robotics, computer graphics, and biomechanics to determine the positions and angles of a mechanism's joints, such as in a robotic arm. This method specifically targets positioning the end effector, the terminal component of the mechanism, at a desired location or along a predetermined trajectory. While standard applications involve adjusting joint angles within an existing configuration, inverse kinematic synthesis involves designing new mechanisms to achieve specified motion objectives. The process is challenging due to its nonlinear and combinatorial nature, which often leads to multiple viable solutions for a given target. This complexity underscores the importance of inverse kinematics as a field that requires precise, controlled movements and positioning.

**Planar Linkage Mechanisms:** In our study of inverse kinematics, we focus on planar linkage mechanisms, which consist of rigid interconnected links joined by joints that move within a two-dimensional plane (see Fig. 2). These joints include revolute joints, which allow free rotation while keeping the linkage endpoints fixed, and fixed joints, which restrict the linkage end to a stationary position in 2D space. The mechanisms under study have a single degree of freedom, controlled by a rotary actuator connected to a fixed joint. This actuator's movement dictates the entire mechanism's configuration, determining the positions of all other joints and linkages based on its rotation.

**Graph Representation and Optimization Formulation:** We model mechanisms as graphs. Let $M = (J, L)$ represent an undirected graph of a planar linkage mechanism, where $J = \{j_1, j_2, \ldots, j_n\}$ is the set of nodes (joints), and $L = \{(j_i, j_j) | j_i, j_j \in J\}$ is the set of edges (linkages). Each node $j_i$ has a 2D position in space and a binary feature describing its type (fixed or free joint), denoted as $(x_i, y_i, T_i)$, where $x_i$ and $y_i$ are the positions of joint $i$ in space and $T_i = 1$ for fixed joints and $T_i = 0$ for free joints (see Fig. 2). In our representation, joint $j_0$ is always fixed and joint $j_1$ is always free. There always exists a linkage $(0, 1) \in L$ which connects joints $j_0$ and $j_1$ which is the driven linkage actuated around joint $j_0$ (see Fig. 2). In the specific problem tackled in this paper, the goal is to generate a mechanism of any size ($n \geq 2$) such that the path traced by the final joint matches a given target path (see Fig. 2). This problem can be formulated as an optimization problem:

$$\min_{M, \theta_{start}, \theta_{end}} \quad ||F(M, \theta_{start}, \theta_{end}) - P_{target}||_2$$
$$\text{s.t.} \quad DOF(M) = 1 \tag{1}$$
$$\text{Singularity}(M, \theta_{start}, \theta_{end}) = 0 \quad .$$

Where $M$ is the graph describing the mechanism, and $\theta_{start}$ and $\theta_{end}$ describe the range of motion for the actuator. $F(M)$ is the path traced by the mechanism $M$, and $P_{target}$ is the target path. $DOF(M)$ is the degrees of freedom for mechanism $M$, which has to be always 1 (to be exactly defined by the motion of the single actuator), and $\text{Singularity}(M, \theta_{start}, \theta_{end})$ is a function that determines if there are any singularities in a given mechanism $M$ for the actuator range of motion from $\theta_{start}$ to $\theta_{end}$.

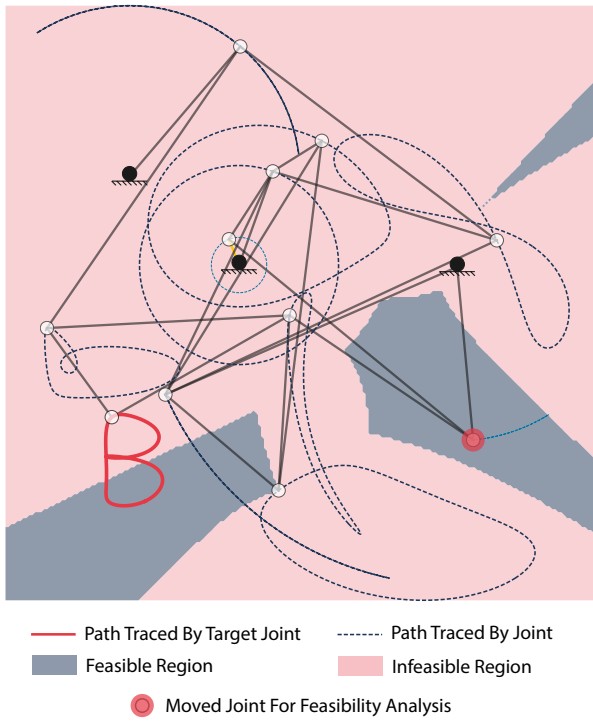

Figure 3: This figure shows the mechanism that traces B and the feasible region highlighted in blue, where the highlighted joint can move without causing a singularity. This shows the nonlinear and discontinuous nature of the problem and highlights the challenge of performing kinematic synthesis. This is only assuming all other joints keep the same positions. Changing the position of any joint could change the feasible region for all other joints making the problem very challenging from the perspective of conventional optimization.

**Singularity Constraints:** Singularities occur when the mechanism $M$ cannot be solved, leading to a locked state where the actuator cannot complete its intended range of motion (as shown in Figure 3 with the red region). The challenge arises in defining the mechanism $M$, which includes specifying the number of joints (integer), their positions (continuous), their types (binary), and the linkage arrangement $L$. This arrangement $L$ can be represented by the adjacency matrix of $M$, containing $\frac{(n^2-n)}{2}$ binary values for $n$ joints. The problem is nonlinear and involves different variable types. Further, singularities are all nonlinear and non-differentiable with respect to all or some of the optimization variables, and graph structure adds combinatorial complexity. This complexity makes traditional optimization infeasible for large $n$, offering an opportunity for deep learning to address these difficult problems.

## 2.2 Computational Design of Planar Linkage Mechanisms

This section outlines significant research efforts aimed at solving inverse kinematics problems similar to the one discussed in this paper. The literature on path synthesis methods falls into three main categories: numerical atlas-based, optimization-based, and deep learning-based approaches. Detailed discussions are provided in Appendix A.

**Numerical Atlas-Based Approaches:** These involve creating a database of mechanisms and their paths, which serves as a "numerical atlas" to find the best match for a target path. However, comparing targets to large databases can be computationally expensive, so these databases are often limited in size or to specific mechanism types, like four-bar or six-bar mechanisms McGarva (1994); Chu & Sun (2010b); Sun et al. (2015).

**Optimization-Based Approaches:** These predominately involve modified versions of common conventional algorithms. Methods include genetic algorithms Lipson (2008); Khan et al. (2015b), Fourier descriptor-based optimization Ullah & Kota (1997); Wu et al. (2011), and Mixed-Integer Conic/Nonlinear Programming (MICP, MINLP) Pan et al. (2023). The approaches often refine existing solutions Bächer et al. (2015); Thomaszewski et al. (2014) or are limited to specific problems Lipson (2008); Baskar & Plecnik (2021). Despite their generalizability, the effectiveness of these methods is limited by their computational intensity and they often struggle (as our results show) to work for large mechanisms with many joints and target points due to the highly constrained nature of the problem.

**Deep Learning-Based Approaches:** These aim to accelerate and enhance path synthesis by integrating traditional methods into data-driven frameworks. The techniques involve using generative models such as Variational Autoencoders (VAEs)Kingma & Welling (2014), clustering-based searchesDeshpande & Purwar (2019a;b; 2020), and Reinforcement Learning (RL) Fogelson et al. (2023). While promising, these methods either require retraining (RL-based methods) for new targets or can be limited by dataset size and mechanism type Deshpande & Purwar (2021); Vasiliu & Yannou (2001).

In this paper, we propose a hybrid method that combines the speed of deep learning with the precision of optimization, aiming to surpass current state-of-the-art methods in both inference time and performance.

## 2.3 Contrastive Learning and Retrieval

Retrieval, in the context of machine learning, refers to the process of finding and returning relevant information from a dataset in response to a query. This can involve matching query features with items's features in the dataset to identify the best matches based on similarity metrics. While most deep learning retrieval research has focused on vision applications like Bontent-Based Image Retrieval (CIBR) Dubey (2022); Chen et al. (2023), our work shifts attention to cross-modal retrieval, particularly retrieving graphs of mechanisms based on their 2D kinematic paths. See Appendix B for more details.

Most cross-modal works involve retrieving images based on multimodal information, such as text and other images. Early works like the correspondence autoencoder (Corr-AE) Feng et al. (2014) train autoencoders for text and images to create embeddings close to each other. The Deep Visual-Semantic Hashing (DVSH) model Cao et al. (2016) and similar methods like Textual-Visual Deep Binaries (TVDB) Shen et al. (2017) create joint embedding spaces for text-based retrieval. Adversarial learning approaches, such as Self-Supervised

Adversarial Hashing (SSAH) Li et al. (2018), use adversarial networks to learn features and hash codes for different modalities Wang et al. (2017); Zhang et al. (2018); Gu et al. (2019); Ji et al. (2019). These methods primarily specializing in text and image. This makes them less adaptable to general cross-modal tasks. Our work requires methods robust to general cross-modal retrieval, leading us to contrastive learning-based retrieval.

Contrastive learning is a technique in machine learning where the model learns to distinguish between similar (positive) and dissimilar (negative) pairs of data points Jaiswal et al. (2021). This method is often used in unsupervised or self-supervised settings to effectively encode information about data without requiring explicit labels for every example. It focuses on pulling representations of similar items closer and pushing representations of dissimilar items farther apart in the embedding space. Seminal works in this field have demonstrated its effectiveness in both supervised and unsupervised tasks Hjelm et al. (2018); van den Oord et al. (2019); Chen et al. (2020a), with foundational works like SimCLR Chen et al. (2020a;b) establishing a general framework for contrastive learning. This inspired further developments such as the Contrastive Language-Image Pretraining (CLIP) model, which generates a cross-modal embedding space for text and images through a generalized loss function:

$$\mathcal{L}_{\text{CLIP}} = -\frac{1}{N} \sum_{i=1}^{N} -\log \frac{\exp(\text{sim}(f_i, g_i)/\tau)}{\sum_{j=1}^{N} \exp(\text{sim}(f_i, g_j)/\tau)}, \tag{2}$$

where $\text{sim}(f_i, g_i)$ measures the cosine similarity between feature embeddings $f_i$ and $g_i$, and $\tau$ is the temperature parameter and $N$ is the number of samples (typically batch size during training). This generalizability makes CLIP suitable for downstream tasks, such as multimodal retrieval applications used in our methodology. Shared cross-modal embedding spaces have been shown to be effective for retrieval in such cases. For instance, Izacard et al. (2022) use contrastive embedding spaces for document retrieval. Similar approaches have been explored for tasks like temporal moment retrieval from videos based on text Zhang et al. (2021) and text-based molecular retrieval for molecule design Liu et al. (2023). Contrastive learning has immense potential for simultaneous modeling of design and performance spaces in engineering applications. For the current problem, we create a contrastive learning-enabled cross-modal approach for retrieving mechanisms based on their kinematic paths. More details are provided in subsequent sections.

## 3 Methodology

Our methodology is illustrated in Figure 1, where we begin by retrieving potential matches from a dataset of 10 million mechanisms using a processed version of the input curve. These initial matches seed our optimization algorithm to refine the designs further. This section elaborates on our dual approach. Firstly, we train a contrastive learning model that bridges the physics underlying the mechanisms with their design representations. This model is crucial for accurately identifying relevant mechanisms from the extensive dataset. Secondly, we detail our optimization algorithm, termed the Learning-accelerated Inverse Kinematics (LInK) algorithm, which builds upon the foundations laid by the contrastive retrieval model. This 'deep optimization' strategy enhances our capability to fine-tune and improve the designs based on the initial matches identified by the retrieval process.

### 3.1 Contrastive Learning Framework

The first stage of our framework involves retrieving candidates from a massive dataset of linkage mechanisms for a given target curve such that these candidates are best suited to be refined for the given input. We use the LINKS dataset proposed by Heyrani Nobari et al. (2022), which includes 100 million mechanisms with up to 20 joints. Searching through such a massive dataset of this size by comparing curves and using conventional methods would be prohibitively expensive and time-consuming. Furthermore, most conventional metrics for searching in such datasets are sensitive to rotations, scale, and other spatial transformations, which makes them less useful. To improve this in our approach, we build a cross-modal contrastive embedding to quickly

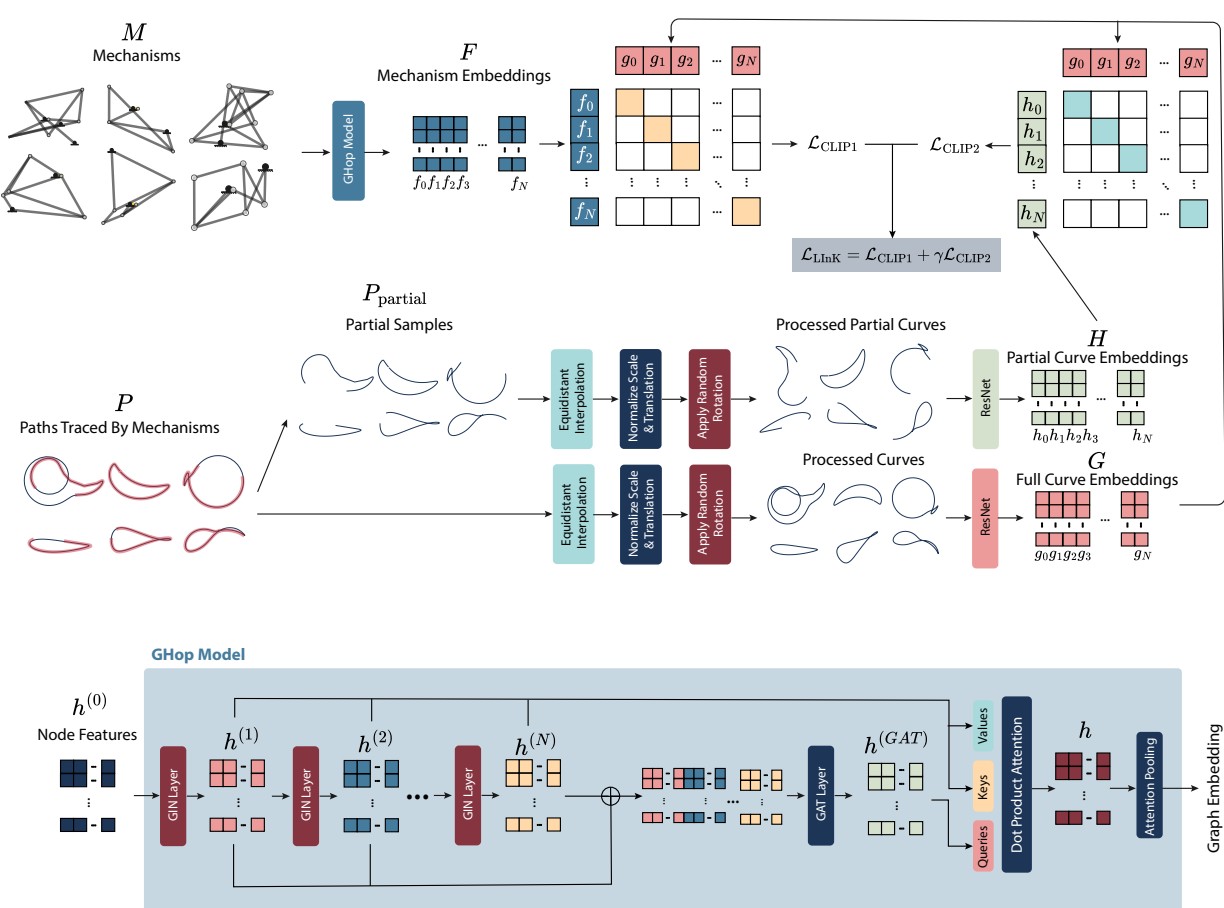

Figure 4: Overview of the cross-modal contrastive learning approach in LInK. Three different representations across multiple modalities are brought into the same embedding space using contrastive learning. The bottom figure also demonstrates the GHop architecture for hop attention that enables us to capture the kinematics of mechanisms.

search through a massive dataset in practical time. In this section, we describe the details of how we built this model and how we build robustness to different spatial and geometric transformations.

### 3.1.1 Mechanism Representation and Model

We employ a subset of 10 million mechanisms from the LINKS dataset, which undergoes extensive pre-processing for normalization. To achieve scale invariance, we standardize the actuator arm length to 0.05 units. We further normalize these mechanisms with respect to translation by centering the actuators at the origin. Additionally, the mechanisms are aligned such that the actuator angle is set to zero, standardizing both position and rotation. This normalization process is critical for the effective training of our contrastive learning model. Furthermore, we meticulously process the dataset to eliminate any redundant joints, i.e., a joint in the mechanism whose kinematic solution depends on the kinematics of all the other joints.

In our approach, we model planar linkage mechanisms as undirected graphs $M$ as we described above with slight variation. The only difference between the graphs we use for training and the prior described graphs is that instead of only two joint types and initial positions, the node features of the graph are defined such that the first element row indicates if a node is fixed or moving (0 for moving joints and 1 for fixed joints), the second element indicates whether a joint is actuated and the third indicates if a given joint is the terminal joint (i.e., the joint which should produce the target curve). The two other elements in each row are set to be

the initial positions of the nodes at time zero. In training our model, we employ a sophisticated multi-layer, multi-level architecture inspired by the Graph Isomorphism Network (GIN) approach proposed by Xu et al. (2019). Off-the-shelf graph models proved inadequate due to the intricate relationship between mechanism configurations and their kinematics, which standard message-passing techniques struggle to capture. To address this, we devised a novel architecture which we term Graph Isomorphism Hop-Attention (GHop), inspired by mechanism solvers. Unlike traditional GNNs where message passing occurs uniformly across all layers, GHop allows for adaptive processing of nodes. This is particularly relevant in mechanism synthesis, where nodes are only processed until they are solved, as illustrated in Figure 5. GHop enables the model to adapt the receptive field for each node separately through an attention mechanism, allowing it to incorporate this feature of linkage mechanisms and their solutions in an unsupervised way. This architecture enables both shallow and deep message passing within the same graph structure, optimizing computational efficiency and improving the representation of linkages compared to baseline GNN models. Our approach modifies the traditional GIN computation as follows:

$$h_v^{(k)} = \text{MLP}^{(k)} \left( \left( 1 + \epsilon^{(k)} \right) \cdot h_v^{(k-1)} + \sum_{u \in \mathcal{N}(v)} h_u^{(k-1)} \right), \tag{3}$$

where $h_v^{(k)}$ is the output features for node/joint $v$ at layer $k$, $\epsilon^{(k)}$ is a learnable parameter as described in the original work by Xu et al. (2019) and $\mathcal{N}(v)$ refers to the set of neighbors of node $v$. In our approach, we introduce hop-attention, where at the end of computing all layers we compute a final feature for each node through a graph attention (GAT) driven attention mechanism. To do this, we take the outputs of all layers, concatenate them for each node, and perform a single GAT convolution on the resulting graph:

$$h_v^{(GAT)} = \overset{n=N}{\underset{n=1}{\|}} \sigma \left( \sum_{u \in \mathcal{N}(v)} \alpha_{vu}^n \mathbf{W}^n \overset{k=K}{\underset{k=1}{\|}} h_v^{(k)} \right), \tag{4}$$

where $h_v^{(GAT)}$ is the output of the graph attention convolution, and $N$ is the number of attention heads in the GAT. $W^n$ is the weights of each attention head linear transformation, while $\sigma$ is a nonlinearity function, and $h_v^{(k)}$ is the output of the $k$-th layer of the GIN model. We then use the output of the GAT as the query. We additionally use the outputs of each layer for a given node as the keys and values for a multi-headed dot-product attention. The attention is applied to obtain the final output of each layer. This means that at the end of each node/joint in the graph/mechanism, we have the following:

$$h_v = \left( \overset{i=N}{\underset{i=1}{\|}} \text{softmax} \left( \frac{\mathbf{W}_i^Q \left[ h_v^{(GAT)} \right] \left( \mathbf{W}_i^K \left[ h_v^{(1)} h_v^{(2)} \cdots h_v^{(M)} \right]^T \right)^T}{\sqrt{d_k}} \right) \left( \mathbf{W}_i^V \left[ h_v^{(1)} h_v^{(2)} \cdots h_v^{(M)} \right]^T \right) \right) \mathbf{W}^O, \tag{5}$$

where $h_v$ is the final output for node/joint $v$ and $N$ is the number of attention heads. $\left[ h_v^{(GAT)} \right]$ is a single row matrix with $h_v^{(GAT)}$ as its only row, and $\left[ h_v^{(1)} h_v^{(2)} \cdots h_v^{(M)} \right]$ is a matrix whose columns are assembled by the output of each layer of the GIN model. $\mathbf{W}_i^Q, \mathbf{W}_i^K, \mathbf{W}_i^V$ refer to the weight of the linear transformation applied at each attention head, and $\mathbf{W}^O$ is the final linear transformation applied to the concatenated attention head outputs resulting the final representation for each node/joint. The intuition behind this kind of architecture is evident in the solver's approach to computing the kinematics of the mechanisms. In the solver we start with the known joints (i.e., 0-1 hops), then propagate the solution to a joint with two known neighbors ($1^{st}$ hop) and repeat this process until all joints are solved (variable number of hops depending on the mechanism skeleton). This means to correctly describe the kinematics of the mechanism each joint requires a different number of hops (see Figure 5). Conventional graph convolution approaches simply do not

account for this, as the nature of the problems they encounter are not like the ones we have in our problem. Given this limitation, it is intuitive to add a mechanism to account for the variable number of hops needed for each joint. Our GHop approach enables this through a simple yet effective attention mechanism. One thing to note is the hop attention mechanism (HAT) is independent of the graph convolution mechanism used and any graph convolution or message passing approach can be used with HAT, which should enable improvement in existing models to solve problems that involve similarly hop-count sensitive challenges (see Figure 4 for details on the architecture).

The final stage of our methodology involves employing an 8-layer model and a hidden size of 768 to compute the final features for each node and take an attention-pooled embedding for each mechanism as the overall mechanism embedding.

### 3.1.2 Path Representation and Building Robust Invariances for Retrieval

In our approach to mechanism retrieval, the goal is to identify mechanisms that accurately trace a target curve without concern for the curve's scale, location, orientation, or the tracing velocity at different parts of its motion. To achieve invariance to these factors, we employ specific preprocessing steps:

- **Normalization for Scale and Translation**: We utilize the first two steps of Procrustes analysis to normalize the scale and translation of the curves. This is done by centering the curve around its mean and scaling it by its standard deviation:

$$X_{\text{normalized}} = \frac{X - \overline{X}}{\sqrt{\frac{\sum_{i=1}^{N}(X_i^1 - \overline{X^1})^2 + (X_i^2 - \overline{X^2})^2}{N}}}, \tag{6}$$

  Where $X$ is the curve being normalized represented as a $N \times 2$ matrix, and $\overline{X}$ is the average value across each column of $X$, and $X_i^1$ refers to the x value of the $i$-th row of $X$, and $X_i^2$ refers to the y value of the same row of $X$.

- **Building Rotation Invariance**: To build invariance to rotation, we randomly rotate curves during training.

- **Timing Invariance**: We also need to build invariance to timing; i.e., the curves should be made up of points that are equidistant from one another. To do this, we simply take any given curve and interpolate points across the curves such that the points are equidistant. Being invariant to timing and making paths equidistant is important for path synthesis since in this kind of problem we do not care about the speed and timing of the motion rather just the shape (see Appendix C for more details).

These steps, illustrated in Figure 4, ensure that our contrastive learning model becomes robust to irrelevant geometric transformations, focusing purely on the kinematic compatibility of the mechanisms with the target curves.

**Modeling Partial Target Curves:** The curves in the LINKS dataset are always closed. However, the target curves that one can give to a model during testing may not always be closed. To effectively manage both closed and open target curves in our contrastive cross-modal space, we implement a robust preprocessing strategy that accommodates varying curve types. To do this, we randomly generate partial curves of the original curves that a mechanism generates, normalize them exactly like we do for the full curves, and remove timing from them (see Figure 4). To gather these samples we randomly cut out a portions of the full closed curves (between 10% to 100% of the curve) and process them as described before. We then introduce partial curves as a separate modality into the overall contrastive learning space to ensure that open curves are also handled by the contrastive representation space. This is a unique contribution of this approach, as previous approaches are limited to closed curves. The ability to handle partial curves is particularly challenging

and important. Unlike closed curves, partial curves lack a clear correspondence with the full mechanism-generated curves. This makes traditional matching techniques, such as Procrustes analysis, ineffective as they rely on known correspondences. Moreover, the scale and translation of partial curves relative to the full curves are unknown, further complicating the matching process. By incorporating partial curves into our contrastive learning framework, we enable the model to effectively capture and match these challenging cases, significantly expanding the applicability of our approach to a broader range of real-world scenarios.

### 3.1.3 Consolidated Methodology

In this section, we consolidate our methodology for training contrastive learning models and searching within the dataset. Our approach maps both mechanisms and curves into a unified representation space using contrastive learning. This mapping allows any introduced curve to be accurately paired with corresponding mechanism candidates based on its spatial representation. We employ a dual-model strategy for curves alongside a specialized graph-based model for mechanisms. Specifically, the mechanism model utilizes the previously discussed GHop architecture. For curve processing, we utilize two ResNet50 models (He et al., 2015): one handling full curves and another for partial curves. The training leverages two distinct contrastive losses, akin to the methodology used in the CLIP model (Radford et al., 2021):

$$\mathcal{L}_{\text{LInK}} = \mathcal{L}_{\text{CLIP1}} + \gamma \mathcal{L}_{\text{CLIP2}}, \tag{7}$$

where $\mathcal{L}_{\text{CLIP1}}$ is the loss for the contrastive signal between the mechanism model's output, and $\mathcal{L}_{\text{CLIP2}}$ is the contrastive loss for the partial and full curves models. $\gamma$ is a hyperparameter for applying different weights to these losses.

The individual CLIP-based loss terms, $\mathcal{L}_{\text{CLIP1}}$ and $\mathcal{L}_{\text{CLIP2}}$, are computed as follows:

$$\mathcal{L}_{\text{CLIP1}} = \frac{1}{N} \sum_{i=1}^{N} -\log \frac{\exp(\text{sim}(f_i, g_i)/\tau)}{\sum_{j=1}^{N} \exp(\text{sim}(f_i, g_j)/\tau)} \tag{8}$$

$$\mathcal{L}_{\text{CLIP2}} = \frac{1}{N} \sum_{i=1}^{N} -\log \frac{\exp(\text{sim}(g_i, h_i)/\tau)}{\sum_{j=1}^{N} \exp(\text{sim}(g_i, h_j)/\tau)}, \tag{9}$$

where $\text{sim}(a, b)$ measures the cosine similarity between feature embeddings $a$ and $b$ extracted by any of the models, and $\tau$ is the temperature parameter controlling the sharpness of the similarity scores. Importantly, the output of the mechanism model corresponds to $f_i$, the output of the full curve model corresponds to $g_i$, and the output for the partial curve model corresponds to $h_i$, and $N$ is the batch size. Notably, mechanisms are matched to their corresponding curves, and presumed dissimilar to others in the batch. While this assumption holds given our diverse dataset, no specific measures have been implemented to handle potential similarities between different mechanisms' curves. Handling partial curves as a separate modality is crucial due to their inherently noisier signal. Partial curves often contain common repeated patterns, such as arcs, which can lead to ambiguous associations if directly matched with mechanisms. By treating them separately and matching them with full curves, we leverage more detailed geometric information like curvature, allowing for cleaner associations. This approach helps maintain the quality of the learned representations and prevents the introduction of noise into the mechanism-curve associations. This setup is visualized in Figure 4, detailing the interaction between these components in our training framework.

Once these models are trained, we have effectively established an approach to map mechanisms (design space) and curves (performance space) into a unified space. This mapping enables rapid searches across our extensive dataset for any given curve by leveraging precomputed embeddings of all mechanisms. For retrieval, we compute cosine similarities between the target curve's embedding and those of the mechanisms, subsequently ranking them by similarity.

### 3.2 Path Synthesis Framework

Our framework for path synthesis includes three stages.

1. **Initial Search**: In the first stage, we perform a search on a 10 million sample subset of the LINKS dataset and find an initial pool of candidates based on the input curves using a contrastive learning-based search.

2. **Batch Optimization**: In the second stage, we perform a gradient-based optimization building off of the BFGS BROYDEN (1970); Fletcher (1970); Goldfarb (1970); Shanno (1970) method, which is applied to all of the mechanisms retrieved in the first stage. To accelerate the process, we implement a batch BFGS optimization on GPU which performs the BFGS optimization on multiple mechanisms simultaneously. We describe this in more detail in this section. In this stage we only perform 10 steps of optimization as optimizing the larger initial pool of candidates can be slow.

3. **Final Refinement**: After the batch optimization of the candidates, we then re-evaluate the mechanisms and pick the top 10% amongst the candidates to move to the next stage. In the final stage of optimization, we further refine the smaller candidate pool using the BFGS optimization for 150 steps, at which point the final design is output to the user.

Each step of this framework is crafted to ensure a fast yet accurate synthesis of paths, optimizing performance and computational efficiency. The following sections will delve deeper into the specifics of these optimization techniques.

#### 3.2.1 GPU-Accelerated Batch Simulation of Linkage Mechanisms and Differentiation

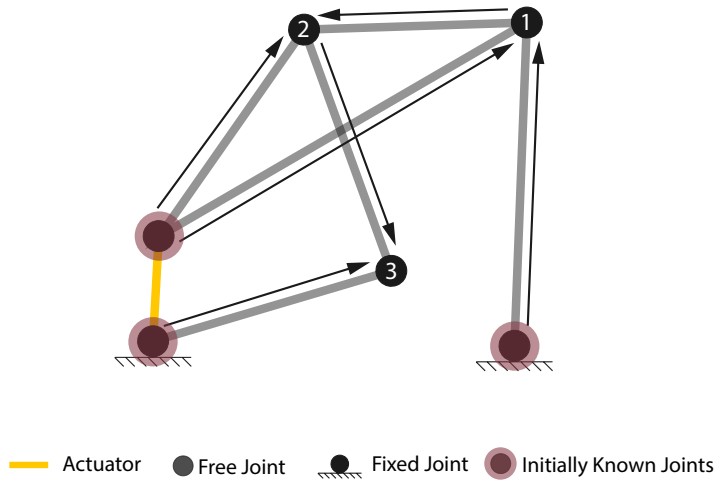

Figure 5: This figure illustrates the path the solver takes to solve the kinematics of a given mechanism, showing how the skeleton of a mechanism (combinatorial design variables) is involved in the mechanism solution. Initially, the solver starts with the known joints (i.e., fixed and actuated joints highlighted red), and step by step the solver solves joints with two solved neighbors (Eqn. 10).In this example, joint 3 (the last joint to be solved) is solved in three steps. The numbered joints indicate the order of solution.

In our gradient-based optimization step, we focus only on optimizing the positions of joints in mechanisms as connectivity parameters, which determine how joints are linked, are not differentiable. To efficiently conduct these optimizations, we employ a differentiable and vectorized solver that allows for rapid simulation and differentiation, enabling practical batch processing on GPUs.

We write a solver to simulate mechanisms, similar to the one proposed in Heyrani Nobari et al. (2022) for dataset generation. This solver handles the kinematics of mechanisms geometrically. While it is not suitable for the rare instances of mechanisms with complex kinematic loops—a limitation discussed further in the prior work Heyrani Nobari et al. (2022)—it suits our needs. The advantage of this solver is twofold. First, the entire process is differentiable with respect to the initial positions of the joints in a given mechanism and second, the process can be vectorized for a batch of simulations which makes this solver the perfect candidate for a differentiable GPU accelerated framework like ours. Here we will briefly discuss the solver and how we vectorized the solver for batches of simulations.

The operation of this solver involves a process akin to a graph traversal algorithm (See Figure 2). Starting with joints of known positions (such as the actuator arm at a specific angle or fixed joints), the solver iteratively computes the positions of adjacent joints that connect to two already determined positions. This sequence continues until the positions of all joints in the mechanism are established, ensuring efficient computation across batches of mechanism simulations.

To vectorize and parallelize this process we have to perform two steps:

- **Pre-Sorting Mechanisms**: To facilitate vectorization and batch processing, we first establish a consistent solution path for all mechanisms in the dataset. By presorting the joints of each mechanism in a solution-ready order, we eliminate the need to identify the solution path in real time during simulations. This sorting, conveniently, has already been handled by Heyrani Nobari et al. (2022) method.

- **Resizing Mechanisms**: The other necessary step that allows for the batch solution by a vectorized GPU solver is that all of the mechanisms must have the same size (i.e., the same number of joints). To do this, we add unconnected fixed joints to all mechanisms that are smaller than the largest mechanism in the dataset such that the mechanisms in the dataset all have the same number of joints. This step ensures that multiple mechanisms can be processed simultaneously without discrepancies in size.

With the mechanisms appropriately sorted and resized, the next step is to implement the vectorized solver that can efficiently handle batches of mechanisms. This solver operates on the principle of solving for any given joint based on the positions of two known adjacent joints at each timestep using the following equation Bächer et al. (2015):

$$\mathbf{X}_i^T = \mathbf{X}_j^T + ||X_i^T - X_j^T|| \times \mathbf{R}(\phi)\frac{X_k^T - X_j^T}{||X_k^T - X_j^T||}, \tag{10}$$

where $\mathbf{X}_i^T$ is the position of joint $i$ at timestep $T$ and $\mathbf{R}(\phi)$ is the 2D rotation matrix for $\phi$ which is computed as:

$$\phi = \text{Sign}\left[\left(X_{j,y}^0 - X_{i,y}^0\right) \times \left(X_{j,x}^0 - X_{k,x}^0\right) - \left(X_{j,y}^0 - X_{k,y}^0\right) \times \left(X_{j,x}^0 - X_{i,x}^0\right)\right] \times$$
$$\cos^{-1}\frac{||X_j^T - X_k^T||^2 + ||X_i^T - X_j^T||^2 - ||X_i^T - X_k^T||^2}{2 \times ||X_j^T - X_k^T||^2 \times ||X_i^T - X_j^T||}, \tag{11}$$

where $\mathbf{X}_i^0$ is the initial position of joint $i$. As can be seen, these equations are easily vectorizable for multiple mechanisms (essentially by turning the $X$ vectors into tensors with an additional dimension of the same size as the batch size) and multiple time steps. Furthermore, all of the above-mentioned equations are differentiable (except the sign function which will not be differentiable at zero). This allows us to use backpropagation to compute the gradients with respect to the initial positions $X^0$ and run the simulation much faster in a vectorized fashion for a batch of mechanisms and for all timesteps simultaneously. We implement such a GPU-accelerated solver using Pytorch and use the automatic differentiation features of Pytorch to obtain the gradients of the solution with respect to the initial positions of the joints.

To optimize mechanisms for path synthesis we need to use this differentiable solver to compute gradients for an objective function. In our approach, we look at two metrics simultaneously. The first metric we look at is the bi-directional Chamfer distance (CD) defined by:

$$d_{CD}\left(S_1, S_2\right) = \frac{1}{|S_1|} \sum_{x \in S_1} \min_{y \in S_2} \|x - y\|_2 + \frac{1}{|S_2|} \sum_{y \in S_2} \min_{x \in S_1} \|x - y\|_2, \tag{12}$$

where $S_1$ is the set of points in the target curve and $S_2$ is the set of points in the curve traced by a mechanism. Chamfer distance is a well-established point cloud-based shape comparison metric and as we have seen this plays a major role in our approach. Although Chamfer distance provides a great metric for comparing the general shape of the curves, it lacks information about path connectivity or ordering. As such we also have to look at some metric that encompasses this. For this purpose, we use the objective function proposed by Pan et al. (2023) in their MICP optimization approach. This metric is defined as the ordered distance (OD):

$$d_{OD} = \min_{o_1 \in O_1} \frac{2\pi}{N} \sum_{i=1}^{N} \left\| X_{o_1(i)}^{\text{coupler}} - X_i^{\text{target}} \right\|^2 \tag{13}$$

where $N$ is the number of points sampled in both the target curve and mechanism traced curve. $X_i^{\text{coupler}}$ is the $i$-th point on the curve traced by the mechanisms, and $X_i^{\text{target}}$ is the $i$-th point on the target curve. This while $o_1(i)$ is the $i$-th point in the optimal ordering to match the curves and $O_1$ is the set of all possible clockwise and counter-clockwise orderings of the curve traced by a mechanism. The constant $2\pi$ is kept for consistency across different works to allow for meaningful comparison of results to prior works that use this metric. Finally, in our gradient-based optimization, we have the following objective function:

$$d = \gamma_1 \times d_{OD} + \gamma_2 \times d_{CD}. \tag{14}$$

In our approach, we assign weights to each metric in the optimization process, with $\gamma_1 = 0.25$ for Chamfer distance and $\gamma_2 = 1.0$ for ordered distance. Using Chamfer distance alone sometimes results in solutions that disregard the ordering of points, focusing merely on matching point clouds. Incorporating ordered distance helps prevent this issue by maintaining the sequence integrity of points, leading to more accurate path tracing. However, relying solely on ordered distance can yield suboptimal solutions and slow down the convergence rate of the optimization process. Typically, a combination of both metrics tends to produce robust solutions.

### 3.2.2 Batch BFGS Optimization on GPU

The BFGS approach for gradient-based optimization has been established as a robust method for unconstrained nonlinear gradient-based optimization. However, there are two important matters that we must address before being able to apply this method to our problem. First, our problem is not unconstrained. This is because if the initial positions of the mechanism being optimized are moved to a position that leads to the term in the inverse cosine seen in equation 11, becoming greater than 1 or less than -1, we will have a locking/infeasible mechanism. As such, instead of using a simple line search for the optimization steps in BFGS, we use a heuristic approach to prevent infeasible and locking configurations. We thus use the common Wolfe conditions (Wolfe, 1971) for inexact line search to find the step size in each iteration of optimization; however, on top of the Wolfe conditions, we also check that a given step size does not lead to infeasible mechanisms by checking if the simulation has resulted in any NaN values.

The traditional approach of performing BFGS optimization individually for each mechanism is inefficient and time-consuming, particularly when leveraging CPU computation. To address this and take advantage of GPU acceleration, we have developed a new batch BFGS scheme that vectorizes the BFGS Hessian updates and line searches across multiple mechanisms simultaneously. This adjustment allows us to process all optimizations in parallel, significantly speeding up the refinement of candidates and the generation of optimal solutions. For the sake of brevity, further details on how this is done are omitted here; interested readers are referred to our publicly available code for the detailed implementation.

### 3.3 Implementation Details of the Optimization Framework

Our methodology begins by precomputing embeddings for 10 million mechanisms from the LINKS dataset using our contrastive learning model tailored for mechanisms. For any provided target curve, we normalize it as outlined earlier, including a smoothing step for hand-drawn or rough curves. This smoothing involves a fast Fourier transform to retain only the first seven frequencies, which we use to reconstruct the curve and compute the curve's embedding. All further optimization processes use the original, unsmoothed curves. This step is only done for computing the embedding of the target curve; the other optimization steps use the original input curves directly.

Given the embedding for the target curve, we then measure the cosine distance between the target embedding and all of the precomputed embeddings of the mechanisms in our dataset and pick the top 500 most similar (i.e., lowest cosine distance) mechanisms for the optimization. As mentioned earlier, we then perform 10 steps of batch BFGS on all of the 500 mechanisms and the target curves. To do this, we normalize the target curve and the paths generated by the candidate mechanisms, then find the optimal orientation of the target for the curves the candidate mechanisms produce. We then perform a brute-force grid search with 200 equally-spaced rotation angles (from 0 to $2\pi$), and identify the optimal rotation that minimizes the Chamfer distance. This rotation is identified for each mechanism in the candidate set. The target curve is rotated to match the candidate paths as well as possible. Then we perform the BFGS with curves that have been oriented properly. Once the initial 10 steps of BFGS are performed, we then pick the top 50 best-performing at this stage of optimization and perform an additional 150 steps of BFGS on them. We call this *deep optimization* framework for path synthesis the Learning-accelerated Inverse Kinematics (LInK). We look at LInK's performance and efficiency in the following sections.

Please note that the number of candidates and optimization steps at each stage are choices made empirically. We will demonstrate that these choices yield highly accurate solutions in rapid time compared to prior works. This approach strikes a balance between speed and accuracy. The optimization process can be made more computationally intensive if desired, potentially allowing for further improvements in the final solution. However, our current parameters have proven effective in achieving excellent results within reasonable time constraints.

### 3.4 Manufacturability Considerations

So far, we have only focused on the topics of kinematics and path synthesis, however, not all mechanisms that can be kinematically solved can additionally be manufactured. In a mechanism, linkages move freely through space, and therefore are typically stacked in layers to avoid collision. Linkages, or multi-layer joints, that occupy the same space as other linkages must be avoided, which is often resolved by adding more layers to a mechanism, increasing its complexity and decreasing its feasibility. This issue can also be overcome with exotic designs for joints that enable circumventing collisions, but that would be costly and difficult to achieve in practice. As such, we must establish if any given mechanism is manufacturable with conventional parts. We can thus formulate an optimization problem to obtain the optimal configuration for a given linkage mechanism such that the total number of layers needed to manufacture a mechanism is minimized.

Each linkage will have a $z$ value, which we denote with $z_i$ for the $i$-th linkage in the set of linkages $L$. Each joint has an attachment to some number of linkages and collides with a set number of linkages as well. For each joint $j$ in the set of all joints $J$, we denote the set of linkages attached to it as $A_j$, and the set of linkages colliding with the joint as $C_j$ for the $j$-th joint. Finally, each linkage is in collision with some other linkages. We denote the set of linkages colliding with the $i$-th linkage as $O_i$, which excludes self-collision. Finally, a subset of $J$ denoted $G \subset J$ is the set of grounded joints. Our objective is to minimize the maximum of $z_i$ for all linkages. To obtain $O_i$, $A_j$ and $C_j$, we simulate the mechanism and find the colliding geometries. Given

the above notation, we have the following design variables (the set $X$ contains all design variables):

$$
\begin{aligned}
&\text{Design Variables } X: \\
&z_i \quad \forall i \in L \\
&u_j \quad \forall j \in J \\
&v_j \quad \forall j \in J \\
&y_{ji} \in \{0,1\} \quad \forall j \in J - G \text{ and } \forall i \in C_j \\
&x_{ik} \in \{0,1\} \quad \forall i \in L \text{ and } \forall k \in O_i \\
&M \text{ which will represent the maximum value amongst } z_i.
\end{aligned}
\tag{15}
$$

With these design variables and assuming $N$ is a very large constant, we have the following optimization problem:

$$
\begin{aligned}
&\min_{X} M \\
&\text{s.t. } 0 \leq z_i < |L|, \quad \forall i \in L \\
&z_i - z_j + N * x_{ij} \geq 1, \quad \forall i \in L \text{ and } \forall j \in O_i \\
&z_j - z_i + N * (1 - x_{ij}) \geq 1, \quad \forall i \in L \text{ and } \forall j \in O_i \\
&u_j \leq z_i, \forall i \in A_j \\
&v_j \geq z_i, \forall i \in A_j \\
&z_i \geq v_j + 1 - N \times y_{ji}, \quad \forall j \in J - G \text{ and } \forall i \in C_j \\
&z_i \leq u_j - 1 + N \times (1 - y_{ji}), \quad \forall j \in J - G \text{ and } \forall i \in C_j \\
&M \geq z_i, \quad \forall i \in L.
\end{aligned}
\tag{16}
$$

This optimization simulation will yield z values which will be integers ranging from 0 to a maximum of $|L| - 1$, which would be equivalent to having one layer per linkage. It is also important to note that the collisions for fixed joints are not considered. This is because each linkage can be grounded in space without the need for a joint to pass through the mechanism. This formulation lends itself very well to mixed-integer linear programming (MILP) algorithms. In this case, we use Gurobi to solve the problem and determine if any given mechanism yields a feasible MILP problem; and if so, determine what the optimal configuration for that given mechanism is. Note that since all of the operations in our approach happen in batches, at every step, we obtain many candidates, and as such we easily check the manufacturability of each mechanism from best to worst performing until the most manufacturable mechanism is found. This is an important aspect of the process that prior methods have ignored; few check for manufacturability at all. In our work from here on to the end, all mechanisms we identify as solutions are manufacturable; the above MILP is feasible for the mechanisms that are shown and discussed in this paper. For more details on this aspect of the work and visualizations of linkage assemblies please see Appendix D.

## 4  Results and Discussion

In this section, we share the results of a few different sets of experiments using our model and compare the results of our approach to the state of the art. Before discussing the details of the experiment, it is important to first discuss our experimental settings and the evaluation metrics we use to evaluate both our method and other approaches.

### 4.1  Evaluation Metrics and Experiment Details

As we discussed before in path synthesis, the primary aim is to come up with mechanisms that allow for the tracing of a desired target path. As such, to measure the performance of any model for this task we need to look at metrics that allow for comparing two given paths purely based on their shape/path. Here we track two highly correlated metrics but, each provides slightly different insight. These metrics are the Chamfer

distance and ordered distance as discussed in Equation 12 and Equation 13 respectively. We discussed the trade-offs of each metric with regards to comparing paths in Section 3.2.1.

It is also important to note that in our tests we resample all target and traced curves (by the mechanisms) by interpolating the curves to 2000 equidistant points. This is necessary to remove timing bias in the curves traced by mechanisms, since the motion of the mechanism is not at a constant velocity. Removing timing is needed since the objective of path synthesis is to trace a path without any prescribed kinematics such as speed and acceleration. Now that the metrics, and how we measure them are established, we will conduct some tests to compare our method to prior works. Finally, we will introduce two new benchmarks for this problem to establish a concrete benchmark for future works.

## 4.2 Path Synthesis Case Studies

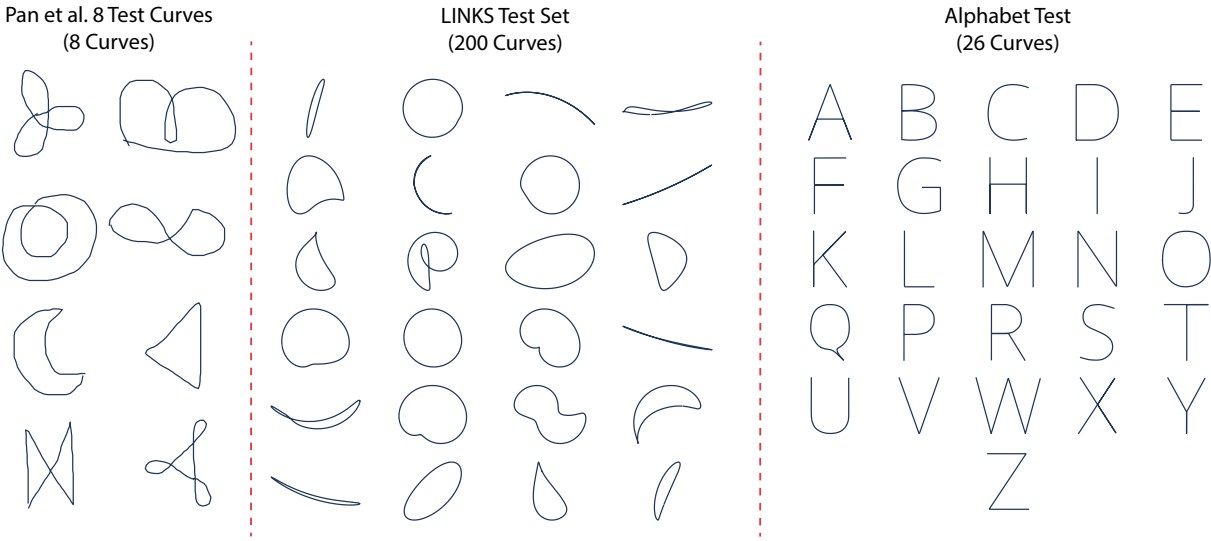

Figure 6: This figure shows the 8 test curves used in prior works (Left), 24 random samples from the LINKS test subset (Middle), and the alphabet test curves (Right).

We will first test our model on three sets of problems and report the results of our approach on each of these test cases. Currently, the existing literature lacks standardized test sets. As such, it is hard for us to compare our model against the state of the art, as these methods have limited code availability and replicating them can be complicated. Despite this, a specific set of test curves have been proposed by Pan et al. (2023), which have been evaluated using multiple methods. However, this set is very small (i.e., only 8 curves), and is not well suited for rigorous testing of path synthesis methods such as ours. However, we use this dataset, as it provides an avenue for comparison to the state of the art.

Table 1: This table presents the quantitative results of our model on the three different test sets we used to evaluate our model. For each test case, we run our model 5 times and report the best solution found within the 5 runs. The numbers following $\pm$ are the standard deviation across the samples in each test case. Note that although we normalize the LINKS and alphabet test curves, we use the original scaling on the 8 test curves, as they were measured in prior works at that scale.

| Test Set | Curve Types | Chamfer Distance | Ordered Distance |
|---|---|---|---|
| LINKS Test Subset | Curves Generated By Random Planar Mechanisms | $0.0059 \pm 0.0029$ | $0.0018 \pm 0.0043$ |
| Popular 8 Curves Test | Hand-Drawn Curves | $0.026 \pm 0.011$ | $0.0106 \pm 0.0083$ |
| Alphabet Benchmark | Curves Mimicking Capital Letters of The English Alphabet | $0.0825 \pm 0.2301$ | $0.179 \pm 0.737$ |

Moving beyond, we also introduce two sets of benchmark problems, each of which provides a different insight into the performance of any method on this problem. First, we use in our approach 200 curves traced by a subset of the LINKS dataset Heyrani Nobari et al. (2022), which we set aside during training for the purpose of testing our method. These curves represent problems that we call *in-distribution*, as these curves are part of the same distribution of data used for training the model. However, it is important to test any method upon curves that are very different from the distribution of the data, to provide an additional challenge by being particularly exotic shapes; curves which planar linkage mechanisms would be ordinarily considered capable of producing. In line with this, we introduce a rather challenging problem: tracing the capital letters in the English alphabet. To do this, we provide 26 continuous curves that align with the shapes of the capital letters in the English alphabet. This set includes shapes that are drastically different from the typical curves in the dataset, hence giving us an *out-of-distribution* test case. Furthermore, these curves are riddled with sharp corners and features that are particularly difficult for planar linkage mechanisms to handle.

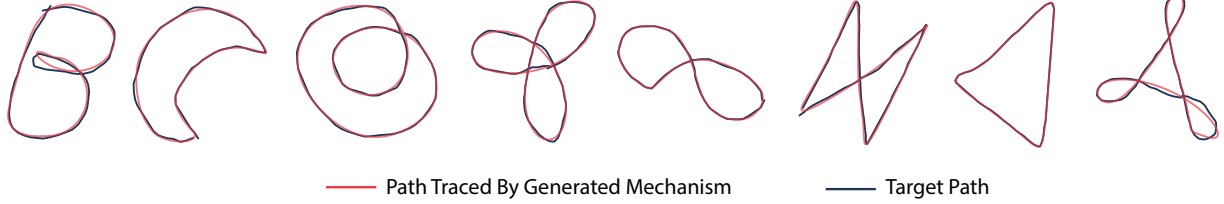

Figure 7: This figure shows the best results from LInK on the 8 test curves used in prior works. The overlayed resulting path traced by the mechanism and the target is shown in red, and the target curve is shown in black. As it is visible here, LInK has matched the target curves effectively without much challenge. For visualization of the underlying mechanisms please see Appendix D.

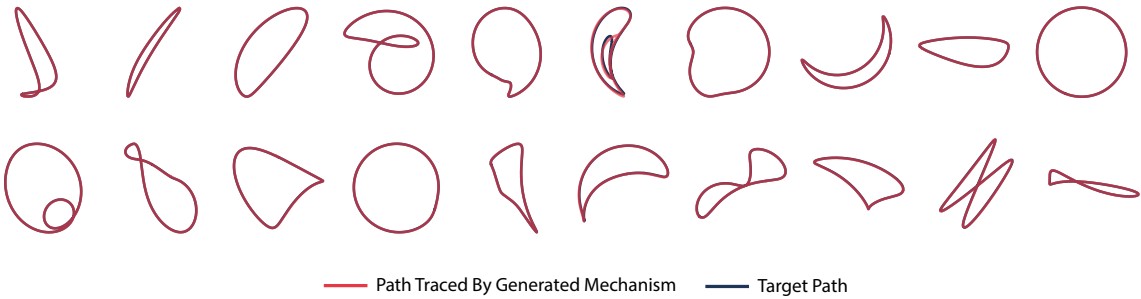

Figure 8: This figure shows 20 random results from LInK on the larger LINKS test subset. We do not visualize mechanisms and only visualize the curve traced by the mechanisms to save space. As it can be seen in almost all cases, the solution found is nearly identical to the target curve, which shows that when it comes to in-distribution target curves, LInK performs exceptionally well.

Each of the three cases discussed above are visualized in Figure 6, which displays all of the curves in the alphabet and the test curves proposed by Pan et al. (2023), while showing a random subset of the dataset test curves. We run our model for all three test cases and display the results visually in figures 9 and 7 for the alphabet and 8 test curves respectively. Finally, we show a few random example solutions from the LINKS test set in Figure 8. As it can be seen when it comes to the 8 test curves our model performs exceptionally well and as we will see later when compared to the state of the art our approach significantly outperforms prior methods. Furthermore, we see that the model's performance on the in-distribution test case is also very good, which should be expected as these samples are similar to the training curves. However, we can see that the curves produced for the alphabet test case do not follow a similar trend. Although for some letters the model has performed well, for many of the complex curves the results are under-whelming and do

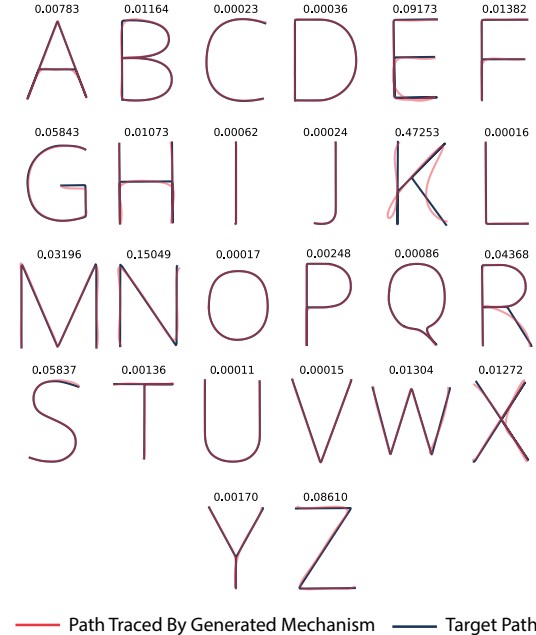

Figure 9: This figure shows results from LInK on the larger alphabet test set. As can be seen, the paths produced by the generated mechanisms do not capture the alphabet shapes as well as we see in the other test sets. This shows that despite LInK performing well even on this test, this test set presents a more challenging benchmark for path synthesis and can serve as a good benchmark for future methods. The ordered distance value for each curve is displayed above the curves for each alphabet path.

not trace the curves as expected. This test set, therefore, serves as a great benchmark for future methods as it presents a more demanding challenge.

Beyond the qualitative results discussed so far, we measure the performance of our model on each of the test sets and report both the ordered distance and Chamfer distance in Table 1. As can be seen, the qualitative observations are confirmed with both metrics, with the model performing well on the in-distribution test and the 8 test curves, while performing notably worse on the challenging alphabet curves (see the ordered distance of the alphabet test).

### 4.3 Comparison To State of Art Methods

In their work, Pan et al. (2023) proposed 8 curves for which they used MICP to perform path synthesis and reported their results. Since then, the current state-of-the-art method in path synthesis using deep learning (i.e., GCP-HOLO Fogelson et al. (2023)), has evaluated their method on this set. As such, we will test our method on this set of target paths as well, and compare our performance to the state of the art based on these paths.

In Table 2, we present the results of our approach for each of the 8 curves: both GCP-HOLO and MICP are used, and as we can see, LInK significantly outperforms these approaches by an order of magnitude demonstrating more than a 94% improvement over the state of the art. One of the main explanations for the shortcomings of the approaches mentioned in this study is that these approaches are limited to smaller mechanisms, due to the high computational cost of dealing with increasingly larger mechanisms with these models. Despite limiting their size, these models are still significantly slower than our proposed framework. We discuss the efficiency and inference speed in later sections. Regardless, we run LInK on limited-sized datasets to demonstrate that LInK outperforms the state of the art, even when limited to smaller mechanisms. To do this we run LInK with a limit of 7 to 20 joints for each curve and show these results in Figure 10. For

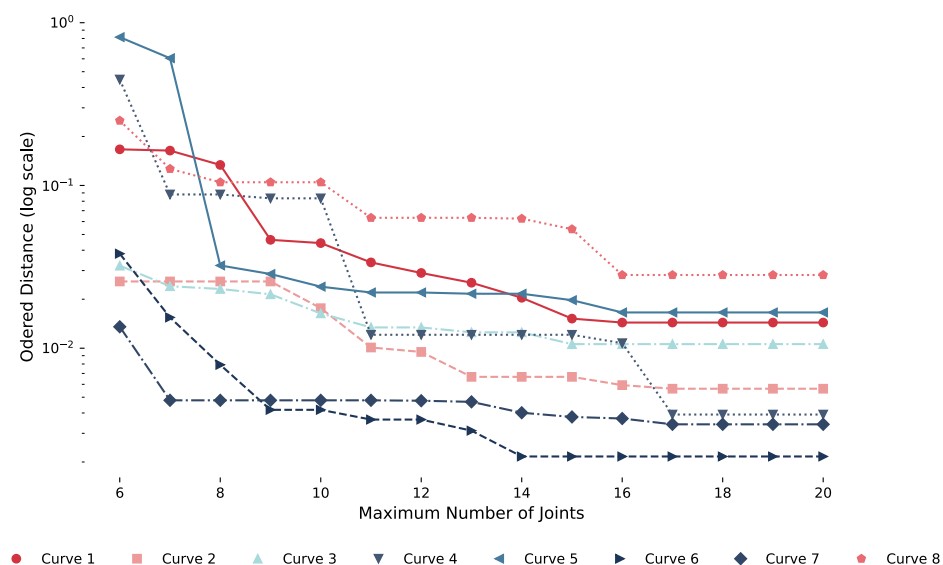

Figure 10: This figure shows how the performance of the LInK method changes with the maximum number of nodes for each curve. As expected, we see a general decrease in error with an increase in complexity, with diminishing returns.

Table 2: This table presents the performance metrics for LInK compared to the state-of-the-art deep learning model, GCP-HOLO, proposed by Fogelson et al. (2023) and optimization-based methods, MINLP and MICP, proposed by Pan et al. (2023). The values reported here are ordered distance in Equation 13. The values for GCP-HOLO, MICP, and MINLP are reported based on the measurements in the original papers Fogelson et al. (2023); Pan et al. (2023). The numbers in brackets after LInK indicate the maximum number of joints LInK was allowed to use (The full LInK method can use up to 20 joints). We observe that on average, LInK has 61 times less error than GCP-HOLO, while being 150 times faster (shown in Table 3)

| Target Curve | LInK | LInK (7) | LInK (11) | LInK (15) | GCP-HOLO | MINLP | MICP |
|---|---|---|---|---|---|---|---|
| | **0.0144** | 0.1638 | 0.0337 | 0.0144 | 2.73 | 0.71 | 2.58 |
| | **0.0056** | 0.0257 | 0.0101 | 0.0059 | 0.32 | 1.25 | 1.37 |
| | **0.0106** | 0.024 | 0.0134 | 0.0106 | 0.44 | 0.70 | 0.77 |
| | **0.0039** | 0.0881 | 0.0121 | 0.0107 | 1.58 | 1.39 | 2.52 |
| | **0.0166** | 0.6052 | 0.022 | 0.0166 | 0.32 | 1.25 | 1.37 |
| | **0.0022** | 0.0155 | 0.0036 | 0.0022 | 1.07 | 0.77 | 1.28 |
| | **0.0034** | 0.0048 | 0.0048 | 0.0037 | 0.87 | 1.36 | 1.36 |
| | **0.0282** | 0.1265 | 0.0633 | 0.0282 | 0.38 | 0.36 | 0.32 |
| Mean | **0.0106** | 0.1317 | 0.0204 | 0.0115 | 0.92 | 0.97 | 1.45 |

these runs, we only conduct the contrastive search on the subset of mechanisms that have fewer than the indicated joints (See LInK(7) through LInK(15) in Table 2).

### 4.4 Inference Speed and Scalability

We have so far only looked at the pure performance of the mentioned methods in terms of how well they can perform path synthesis. However, an important aspect of path synthesis algorithms is their limitations in terms of inference speed and scalability to large, complex mechanisms. To test the scalability and inference speed of different methods, we investigate both the inference time and size of the mechanisms each method can produce.

Table 3: This table shows the average inference time for each method. For SOTA methods we report the rough inference time reported by the authors (they report a range, and we report the middle of that range here), as the code for the work by Pan et al. (2023) is not publicly available. For GCP-HOLO, we run both of their variants on our hardware using an Intel i9-14900K and an RTX 4090 GPU, which provides better inference speed than the authors reported. We also run our model on the same hardware. In each case, 10 tests are run, and the average time is reported below. The percentages are reported based on the inference speed up compared to MINLP. We observe that, on average, LInK is 150 times faster than the fastest GCP-HOLO. At the bottom of the table, we also include the timing of manually searching the dataset of 10M samples for retrieval only using manual ordered distance calculation and using cosine search using for contrastive retrieval. The value of contrastive retrieval is clear with multiple orders of magnitude faster retrieval compared to manual search.

| Model | Maximum of Joints | Solver timesteps | Inference Speed (s) | |
|---|:---:|:---:|---|---|
| MINLP Pan et al. (2023) | 7 | 20 | 36000 | (00.00%) |
| MICPPan et al. (2023) | 7 | 20 | 27000 | (25.00%) |
| GCP-HOLOFogelson et al. (2023) | 11 | 20 | 2475.12 | (93.12%) |
| GCP-HOLO + CMAE-ESFogelson et al. (2023) | 11 | 20 | 2498.23 | (93.06%) |
| **LInK (Ours)** | **20** | **2000** | **15.49** | **(99.95%)** |
| Manual Search of 10M Samples (GPU) | 20 | - | 10178 | |
| Manual Search of 10M Samples (CPU) | 20 | - | 48421 | |
| Contrastive Retrieval In 10M Samples (GPU) | 20 | - | 0.0259 | |
| Contrastive Retrieval In 10M Samples (CPU) | 20 | - | 6.407 | |

Table 3 presents the inference speed and maximum mechanism sizes that each method is capable of producing within that inference time. Furthermore, we show in this table the simulation fidelity, which determines the number of points sampled for each method. We can see that LInK is not only 99.95% faster than the MINLP method, but it is also more than 99% faster than the fastest method in the state of the art – capable of producing mechanisms that go beyond 4-bar and 6-bar mechanisms, GCP-HOLO. Beyond this, we see that LInK is capable of producing much more complex mechanisms with up to 20 joints and simulating with 2000 timesteps during optimization. This is in contrast with the 11 joints and 20 timesteps that the best competing method is working with while having 99% slower inference. We also look at the inference speed of retrieval using manual search (i.e., measuring ordered distance to all curves in the dataset manually and picking the top candidates) and the contrastive retrieval approach we propose. We see clearly that manually searching the dataset would be computationally prohibitive and would require times closer to the slowest baselines of MINLP and MICP, while contrastive retrieval takes under 0.1 seconds accelerating retrieval by many orders of magnitude, while also allowing for retrieval based on partial curves which would be not be as simple as measuring ordered distance. These results demonstrate that frameworks like LInK can be very powerful for speeding up these kinds of optimization-based methods by removing the majority of the burden from the optimizer, enabling faster and more precise optimization.

### 4.5 Validating The Effectiveness of GHop Architecture

To validate the advantage of our GHop architecture over standard GNN architectures, we conducted a comparative performance analysis. We trained the LInK framework using baseline Graph Convolutional

Network (GCN) and Graph Isomorphism Networks (GIN) models, each configured with an equivalent number of layers, hops, and hidden sizes, resulting in approximately the same parameter count: 38M for both GCN and GIN, and 31M for GHop.

We evaluated the effectiveness of each model by monitoring the $\mathcal{L}_{\mathrm{CLIP1}}$ loss on validation data throughout the training period. The results indicated that the GHop architecture consistently achieved lower validation losses compared to the GCN and GIN models, which failed to reduce the loss to comparable levels. This discrepancy in performance was significant, with GHop's validation loss being less than half that of the GCN, and the naive GIN model even showed a higher loss, suggesting a poor capture of the contrastive relationships between mechanisms and curves.

These findings are graphically represented in Figure 11, where GHop's superiority is clearly evident, illustrating its enhanced capability in the LInK framework compared to traditional GNN architectures.

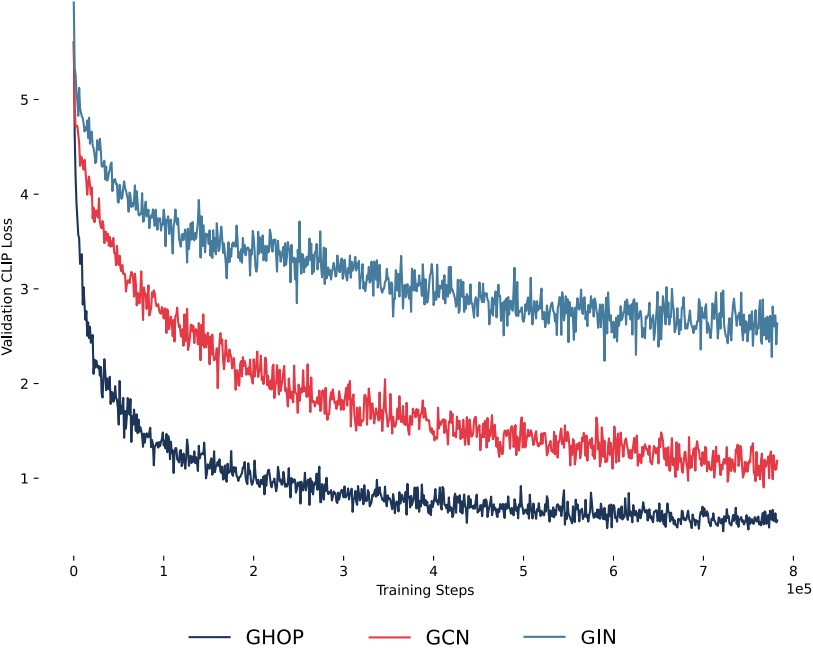

Figure 11: This figure shows how the validation performance ($\mathcal{L}_{\mathrm{CLIP1}}$) of each GNN architecture changes in the LInK training. We see here that the proposed GHop architecture is performing significantly better than alternative GNN architectures.

## 5    Conclusion and Future Work

In this work, we introduced LInK, a novel framework that combines contrastive learning with optimization techniques for effectively solving complex engineering design problems, with a specific focus on the path synthesis problem for planar linkage mechanisms. By leveraging a vast dataset and employing a multimodal, transformation-invariant contrastive learning framework, LInK adeptly captures intricate physics and design representations, enabling the rapid and precise retrieval and optimization of mechanisms. This approach not only significantly expedites the search process, but also enhances precision through a hierarchical optimization algorithm. The results demonstrate the effectiveness of LInK through improvement in both speed and performance by an order of magnitude.

For future work, the adaptability and robustness of LInK opens new avenues for exploring its applicability across a broader spectrum of engineering challenges, beyond linkage synthesis. Direct analogs exist in soft robotics and compliant mechanisms Sun et al. (2024), where kinematic synthesis of a similar nature is investigated in non-rigid body components. In these fields, particularly compliant mechanisms, there often

exists a mixed integer representation of the design space and a curve representation of the performance space with a differentiable solver for continuous variables Sun et al. (2024). Less direct analogs can be found in complex optimization problems such as topology optimization Wang et al. (2021). For instance, in structural topology optimization for minimizing compliance (maximizing stiffness) with complex physics like buckling Ferrari & Sigmund (2019), conventional methods can take hours or days to solve. Our framework could potentially capture such physics in a cross-modal contrastive space, significantly accelerating the process of finding good local candidates. As demonstrated, the integration of contrastive learning-based joint design and performance space representation and optimization has the potential to facilitate advancements in areas where traditional methods falter due to the complexity or combinatorial nature of design spaces. Further research could explore the extension of LInK's methodology to these fields, demonstrating its ability to handle problems that conventional optimization struggles with.

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

## A  Broader Discussion On Computational Kinematic Synthesis of Planar Linkage Mechanisms

As discussed prior most methods in kinematic synthesis fall into three primary categories: numerical atlas-based approaches, otimization-based approaches, and deep learning-based approaches.

**Numerical Atlas:**   In numerical atlas approaches, a database of mechanisms is first created or gathered and simulated to obtain the paths that each mechanism traces in the database. Then these mechanisms and their corresponding paths are used as a *numerical atlas*. Given a target path, one can look up all of the mechanisms in a database to find the best matching candidate in the database McGarva (1994); Chu & Sun (2010b); Sun et al. (2015). It can be computationally expensive to compare a target to a large number of mechanisms; therefore, in the majority of these approaches, the database is usually limited in size or limited to a handful of mechanism types, such as a four-bar or a six-bar mechanism.

**Optimization-Based Kinematic Synthesis:**   A different kind of approach to solving this kind of problem is applying different optimization approaches. Some apply genetic algorithms to the problem Lipson (2008); Khan et al. (2015b), While others have developed Fourier descriptor-based optimization algorithms Ullah & Kota (1997); Wu et al. (2011). At the same time, many approaches focus on specific problems without generalizability or refining existing solutionsLipson (2008); Bächer et al. (2015). Most of the aforementioned methods commonly work with existing solutions Bächer et al. (2015); Thomaszewski et al. (2014) or are limited to specific kinds of mechanisms or problems Lipson (2008); Baskar & Plecnik (2021). Most promising amongst the recent developments is the method proposed by Pan et al. (2023), which formulates the optimization problem discussed prior as a mixed-integer conic-programming (MICP) and mixed-integer nonlinear programming (MINLP). However, these methods use a branch and bounds method to optimize the problem, which limits their capacity in the number of joints and the number of points in their target

curves Pan et al. (2023). Furthermore, even for simple problems, the time to solve any given input is significant (in the order of 5-10 hours) which makes these kinds of methods not only limited by mechanism size and target path point count but prohibitive for generating multiple candidates for the same problem or generating solutions for complex targets that require many points to describe.

**Deep Learning And Kinematic Synthesis:** Given these limitations in the conventional approaches, data-driven and deep learning approaches have been proposed to accelerate and improve path synthesis results. These studies often integrate traditional methods, such as the *numerical atlas* and optimization techniques, into data-driven frameworks. For instance, Deshpande & Purwar (2019a) have refined a method that merges the numerical atlas concept with optimization strategies (Deshpande & Purwar, 2019a;b; 2020), utilizing VAEs (Kingma & Welling, 2014) and a clustering-based search to identify suitable candidates for generating the required coupler curves. Additionally, their subsequent research applies VAEs and conditional VAEs (Sohn et al., 2015) to create mechanisms. However, the datasets employed in these studies are typically small and restricted to certain mechanism types (e.g., four-bar, six-bar, etc.), with one instance being a dataset comprising 6818 linkage mechanisms (Deshpande & Purwar, 2021). Other data-driven studies have explored mechanism generation by conditioning on specific paths (Vasiliu & Yannou, 2001), yet this approach is also mainly limited to four-bar mechanisms. In contrast to these strategies, methods are attempting to apply machine learning to emulate optimization techniques. An example of this is a study employing deep Q learning (Mnih et al., 2013). More recently Fogelson et al. (2023) have introduced a novel reinforcement learning approach that accelerates the process of optimization in comparison to MICP-based approaches mentioned before. Although these reinforcement learning-based strategies are not confined to particular mechanisms, they require retraining for each new target shape, thus limiting their generality and introducing issues similar to the aforementioned MICP-based methods.

It is clear that machine learning techniques hold significant potential in this domain, yet current methodologies exhibit two main shortcomings. Firstly, many techniques are constrained to specific types of mechanisms and problems. Secondly, when these methods are not limited by mechanism type, they mimic optimization and perform the task rather slowly. In our work, we developed a hybrid method that combined the acceleration that is associated with deep learning frameworks and the precision and robustness that is associated with optimization to strike a balance and significantly outperform the state of the art in both inference time and raw performance.

# B    Detailed Review of Deep Learning-Based Retrieval and Contrastive Learning

Here we provide a more in-depth review of the works we discussed in the main body of the paper for further context around the topics of deep learning-based retrieval and contrastive learning.

## B.1    Learning Based Retrieval

When it comes to deep learning-based approaches for retrieval the majority of research has been dedicated to vision applications, specifically Content Based Image Retrieval (CIBR). Dubey (2022) and Chen et al. (2023) have conducted comprehensive surveys on the deep learning developments for different types and applications of CIBR, which provide a more in-depth discussion on the topic for readers seeking a more detailed review. Here we will focus on discussing some of the more directly related works of research on the topic. In our work, we aim to retrieve mechanisms (represented as graphs) given target kinematics (2D paths), which makes the approach one that involves different modalities of data representation. As such here we focus on cross-modal retrieval methods, discuss some of the ways that others have approached problems of this nature, and refer readers to the aforementioned surveys for a review of other types of retrieval models.

Cross-modal image retrieval refers to the task of retrieving images in a dataset based on information involving more than one modality. A prominent example of this is text and image-based retrieval, which comes up in text query-based image retrieval and image labeling based on text retrieval. One of the earliest works on the topic by Feng et al. (2014), called Correspondence AutoEncoder (Corr-AE) trains two autoencoders simultaneously one for text and one for images, and attempts to build correspondence by encouraging the embeddings of the two autoencoders to be close to each other through an L2 loss function between the text and

image embeddings produced by each autoencoder. Cao et al. (2016) introduce a novel Deep Visual-Semantic Hashing (DVSH) model that introduces an end-to-end deep learning architecture called visual-semantic fusion network, which simultaneously captures spatial dependencies in images and temporal dynamics in text. This network is responsible for learning a joint embedding space that mitigates the heterogeneity between the two modalities, thereby generating compact, similarity-preserving hash codes, which can be used for text-based retrieval. Others have looked at similar approaches with slight differences in architecture and implementation such as Textual-Visual Deep Binaries (TVDB) Shen et al. (2017). Many more approaches that have been explored for this kind of cross-modal task with similar overall methodologies Wei et al. (2017); Yang et al. (2017); Wu et al. (2017) with slight nuances, the detailed description of which is outside the scope of this section. Furthermore, some research has been focused on performing the same task through adversarial learning approaches. For example Li et al. (2018) introduces Self-Supervised Adversarial Hashing (SSAH). The methodology combines adversarial learning with a self-supervised semantic network to achieve this goal. Two adversarial networks are employed to learn the high-dimensional features and corresponding hash codes for different modalities. Simultaneously, a self-supervised semantic network, leveraging multi-label annotations, guides the adversarial learning process to maximize semantic relevance and feature distribution consistency between modalities. Again, various works of research have been proposed along the same lines of using adversarial networks Wang et al. (2017); Zhang et al. (2018); Gu et al. (2019); Ji et al. (2019) for this kind of text-to-image or image-to-text retrieval.

The common pattern across the aforementioned works and the majority of works that surround the topic is the fact that these methods all focus primarily on text and image, which means the authors often tailor their approaches and architecture to specialize in this kind of task specifically. This makes these models much better at performing cross-modal retrieval on images and text but makes the approaches very difficult to adapt to general cross-modal retrieval tasks, such as the one we are performing in this work. This means that we must look at methods that are robust to cross-modal retrieval tasks which do not specialize at their core to deal specifically with image and text, and this is where contrastive learning-based retrieval methods come into play.

## B.2 Contrastive Learning and Contrastive Learning Based Retrieval

As discussed before, most of the methods built for retrieval using deep learning were built to be specialized and included mechanisms specific to text and image datasets, which are not adaptable to general tasks like the one we perform here. Given this, and the fact that the inverse kinematic problem involves a very general and context-free nature, the kinematics simulations are the only directly available information for any given mechanism. This exists in isolation, unsupervised, and self-supervised approaches are best suited for building a robust retrieval mechanism for kinematic synthesis. These requirements lend themselves rather well to contrastive learning.

Contrastive learning at its core refers to unsupervised or self-supervised learning methods that aim to develop models that learn to distinguish between similar and dissimilar data points Jaiswal et al. (2021). These methods gained popularity as a result of the early seminal works in deep contrastive learning Hjelm et al. (2018); van den Oord et al. (2019); Chen et al. (2020a) that demonstrated the powerful capabilities of these approaches for both conventionally supervised deep learning problems (e.g. contrastive learning feature extraction for classification and regression) and unsupervised representation learning. Among these early approaches, the work by Chen et al. (2020a;b), known as an effective framework for contrastive learning (SimCLR) stands out as a rather generalizable and robust framework. Although SimCLR itself was built and tested on a single modality of data, primarily images, similar approaches soon emerged in other works of research that adapted similar techniques to contrastive learning on multiple modalities. Most notably the work by Radford et al. (2021), introduces the Contrastive Language-Image Pretraining (CLIP) model, which creates a cross-modal embedding space for text and images. However, unlike highly specialized text-image retrieval models, the CLIP approach is rather generalizable to other cross-modal embedding spaces. This is evident in the loss function used by CLIP (Equation 2). This means that so long as trainable models can be implemented for data representations in each modality, the CLIP approach applies to the problem. Given this robustness and generalizability, we incorporate the CLIP loss in our approach for unsupervised representation learning in the inverse kinematics problem.

The concept of a shared cross-modal embedding space is highly analogous to the retrieval problem discussed before. This is because these models map data from different modalities into the same space based on similarity, which makes these embedding spaces ripe for retrieval problems. As it turns out, this has been explored in many works of research surrounding cross-modal retrieval. In an approach very similar to what we do here Izacard et al. (2022) uses this concept of contrastive embedding spaces for retrieval for the task of document retrieval. These kinds of approaches have also been explored for cross-modal retrieval tasks, such as temporal moment retrieval from videos based on text Zhang et al. (2021) or text-based molecular retrieval for molecule design Liu et al. (2023), which demonstrates the effectiveness of cross-modal embedding spaces built using contrastive learning for retrieval tasks. In our approach, we employ contrastive learning-enabled cross-modal spaces for mechanism retrieval based on kinematic objectives.

## C  Kinematic Synthesis Problems In Planar Linkages

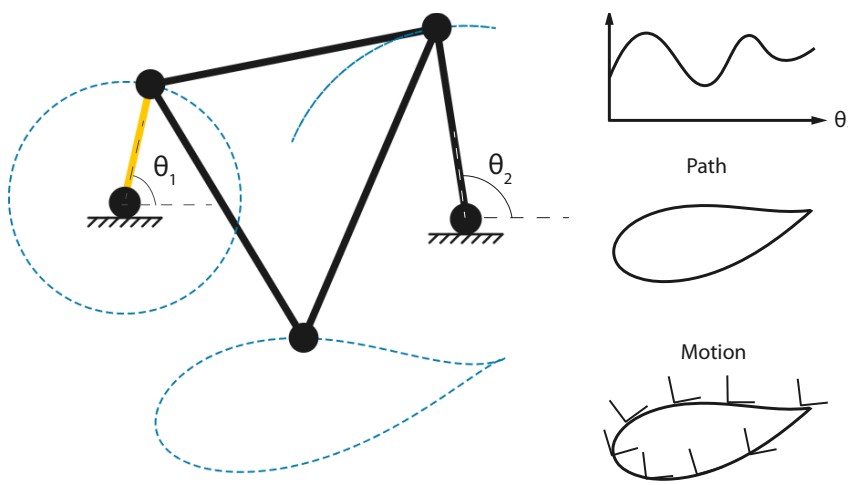

Figure 12: Different types of problems in inverse kinematics of planar linkage mechanisms. Note that the yellow arm is the actuator arm

In these kinds of planar linkage mechanisms, the problem of inverse kinematic synthesis can have different types of objectives, such as path synthesis, motion synthesis, and signal generation McCarthy & Soh (2010). The path synthesis problem can be described as the problem of designing linkage mechanisms that can generate a particular path that is described by a finite series of point coordinates. As such in this type of problem, the speed and timing of the motion of the mechanism are not considered as part of the objective. Motion generation on the other hand can be thought of as the generalized version of the path generation where aside from point coordinates, the orientation of an attached rigid body (such as a robot arm) and its timing and speed are also prescribed. Finally, the function generation problem refers to the problem of generating a specific series of output crank angles (or slider positions) at given angles (or positions) at the actuator, essentially transforming the signal from the actuator (angle or position) to a different signal at the output crank (or slider). See Fig. 12 for more details. LInK is created with a primary focus on the "Path Synthesis" problem, although with some tweaking the data could be adapted to be used for other types of problems such as "Function Generation" and "Motion Synthesis" (See Fig. 12).

## D  Additional Visualizations

Here we will visualize the mechanisms that trace the curves in the 8 test curves and alphabet tests. For each set of tests, we will visualize the kinematics of the entire mechanisms in 2D graph and path figures. We will also provide the optimal assembly of the mechanisms in 3D to visualize what the actual assembled mechanism would look like. Note that these solutions are the best perfoming solutions, as such they tend to have more joints since higher complexity mechanisms tend to produce more precise solutions.

### D.1 Eight Test Curves

Figure 13 shows the mechanisms associated with the solutions found by LInK for each of the 8 test curves. This figure shows the mechanisms and their kinematic in 2D while Figures 14, 15 show the optimal assemblies for each of these mechanisms to avoid collisions while using the fewest layers in the z-axis according to the optimization scheme described in Section 3.4.

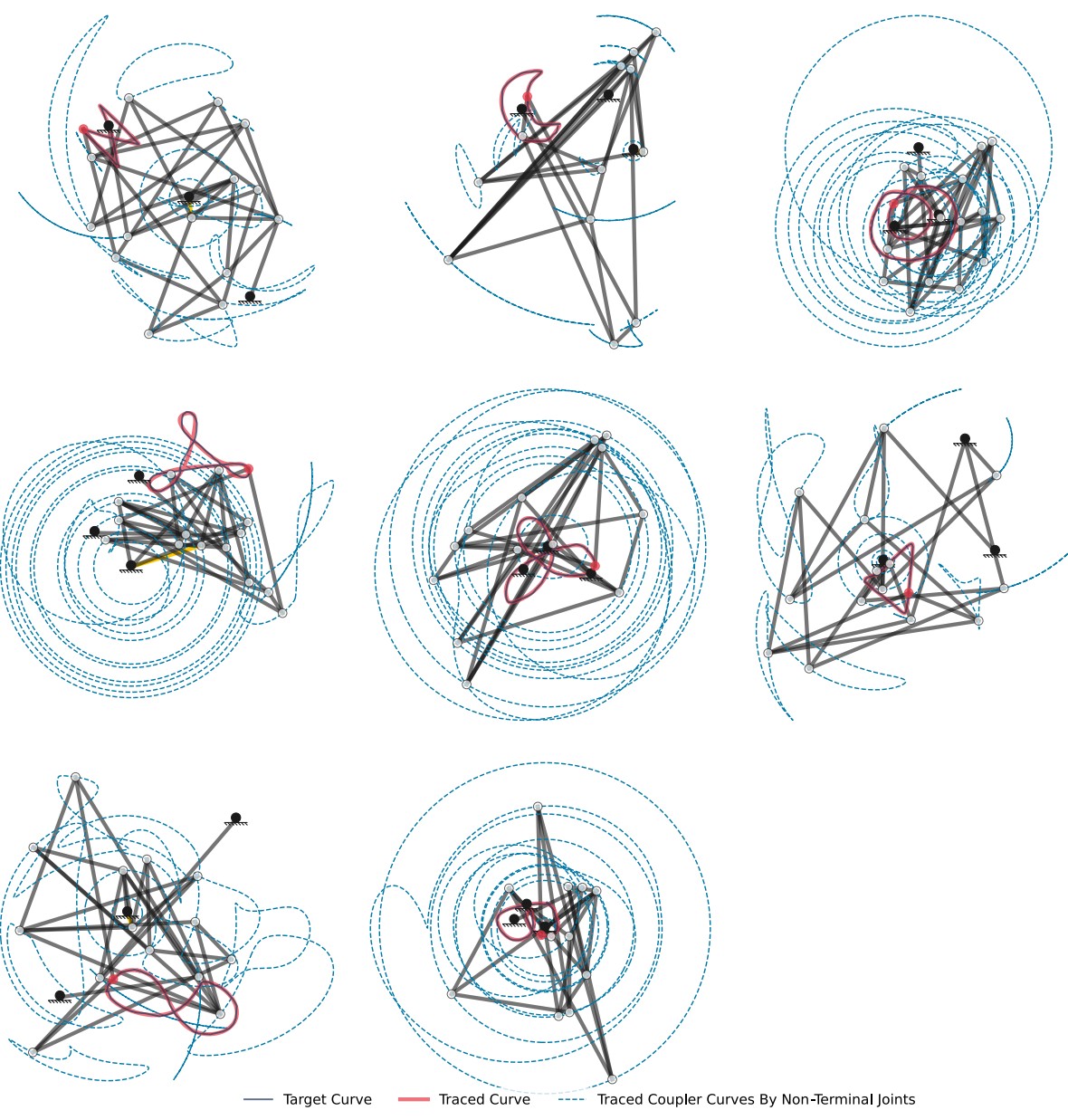

Figure 13: This figure shows the best results from LInK on the 8 test curves. As it is visible here, LInK has matched the target curves effectively without much challenge. Here, we visualize the mechanisms in 2D and also plot the paths traced by all the joints in each mechanism.

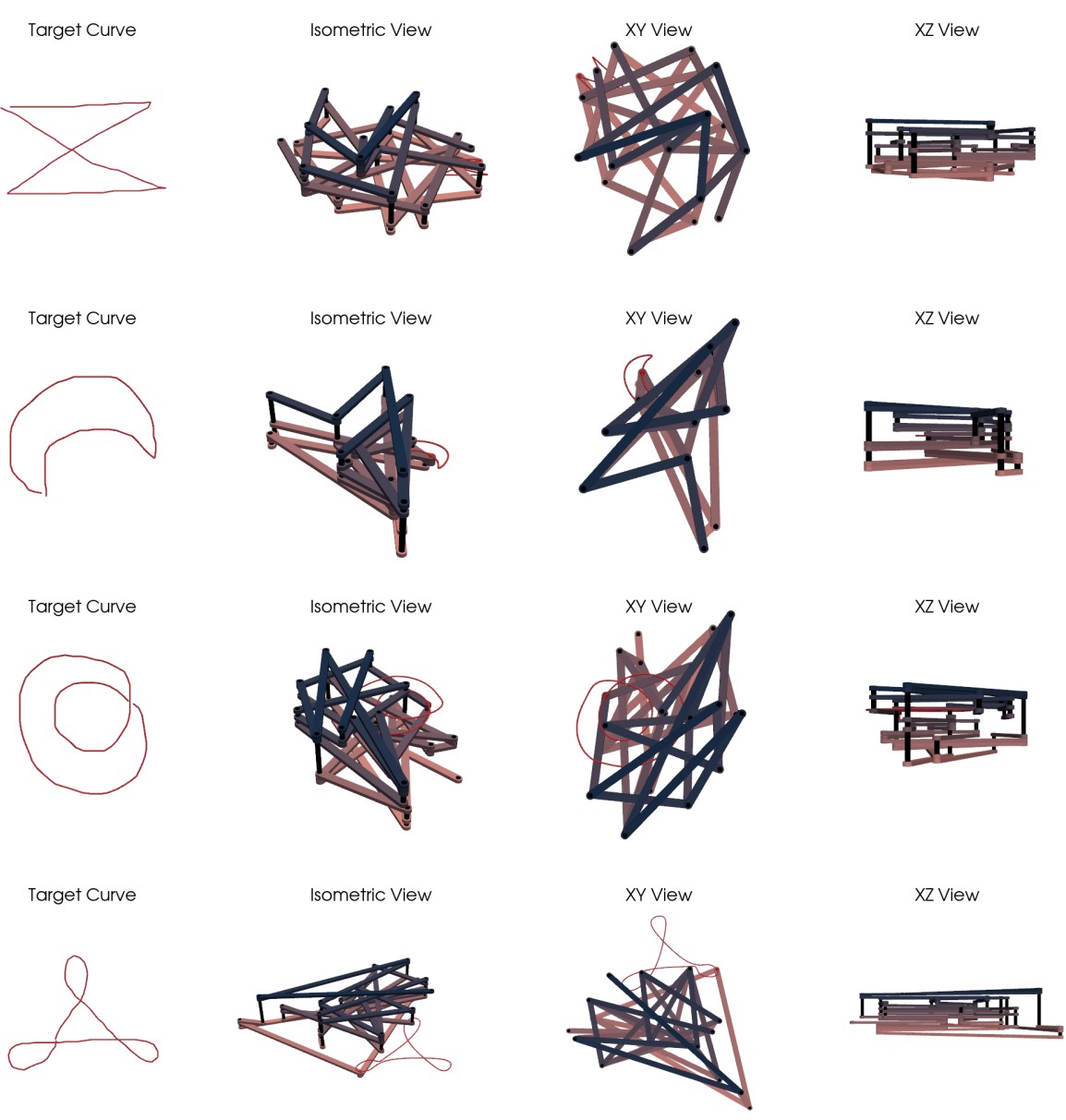

Figure 14: This figure shows the 3D optimal assemblies for mechanisms displayed in Figure 13 (first four curves).

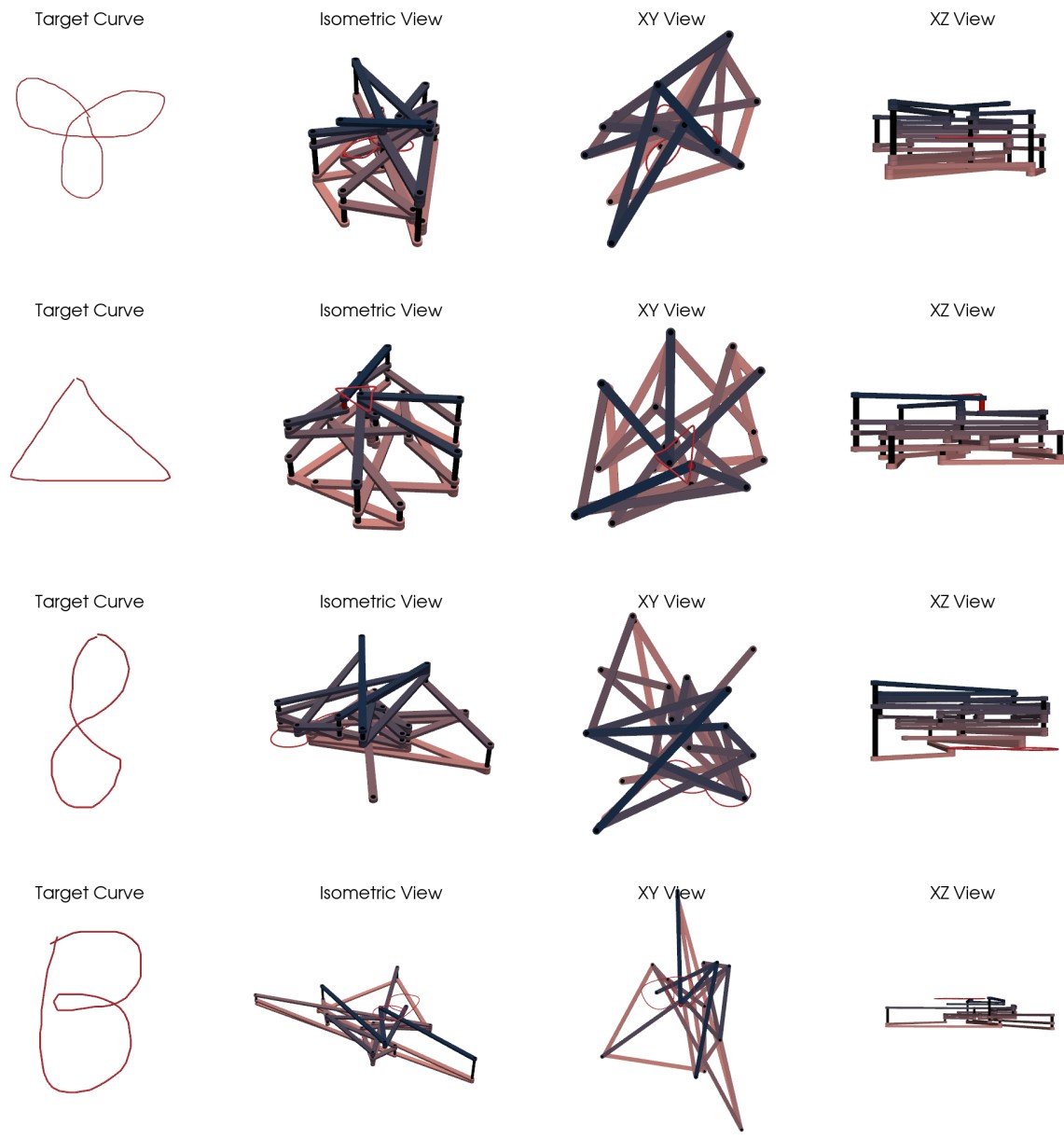

Figure 15: This figure shows the 3D optimal assemblies for mechanisms displayed in Figure 13(last four curves).

## D.2   Alphabet Test Curves

The figures in this subsection are similar in nature to the figures in the prior subsection. Again it is important to acknowledge that these are the best perfoming solutions. Compared to simpler curves –such the letter 'O'– the produced mechanism is more complex because the more complex mechanisms achieve an exceptionally accurate solution. Note that the LInK approach can be limited to any number of joint counts as mentioned in the main text.

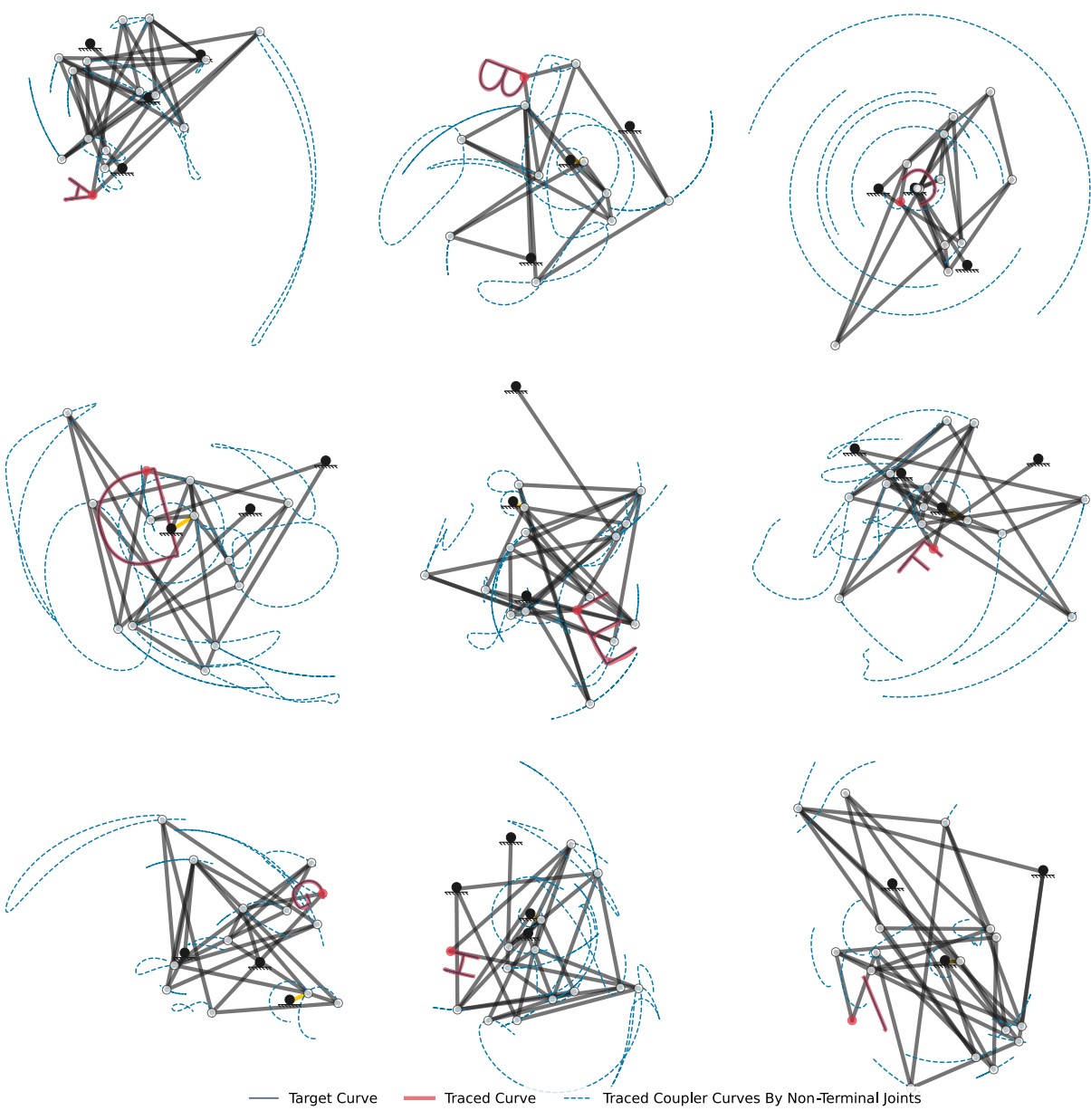

Figure 16: This figure shows the best results from LInK on the first 9 alphabet test curves. As it is visible here, LInK has matched the target curves effectively without much challenge. Here, we visualize the mechanisms in 2D and also plot the paths traced by all the joints in each mechanism.

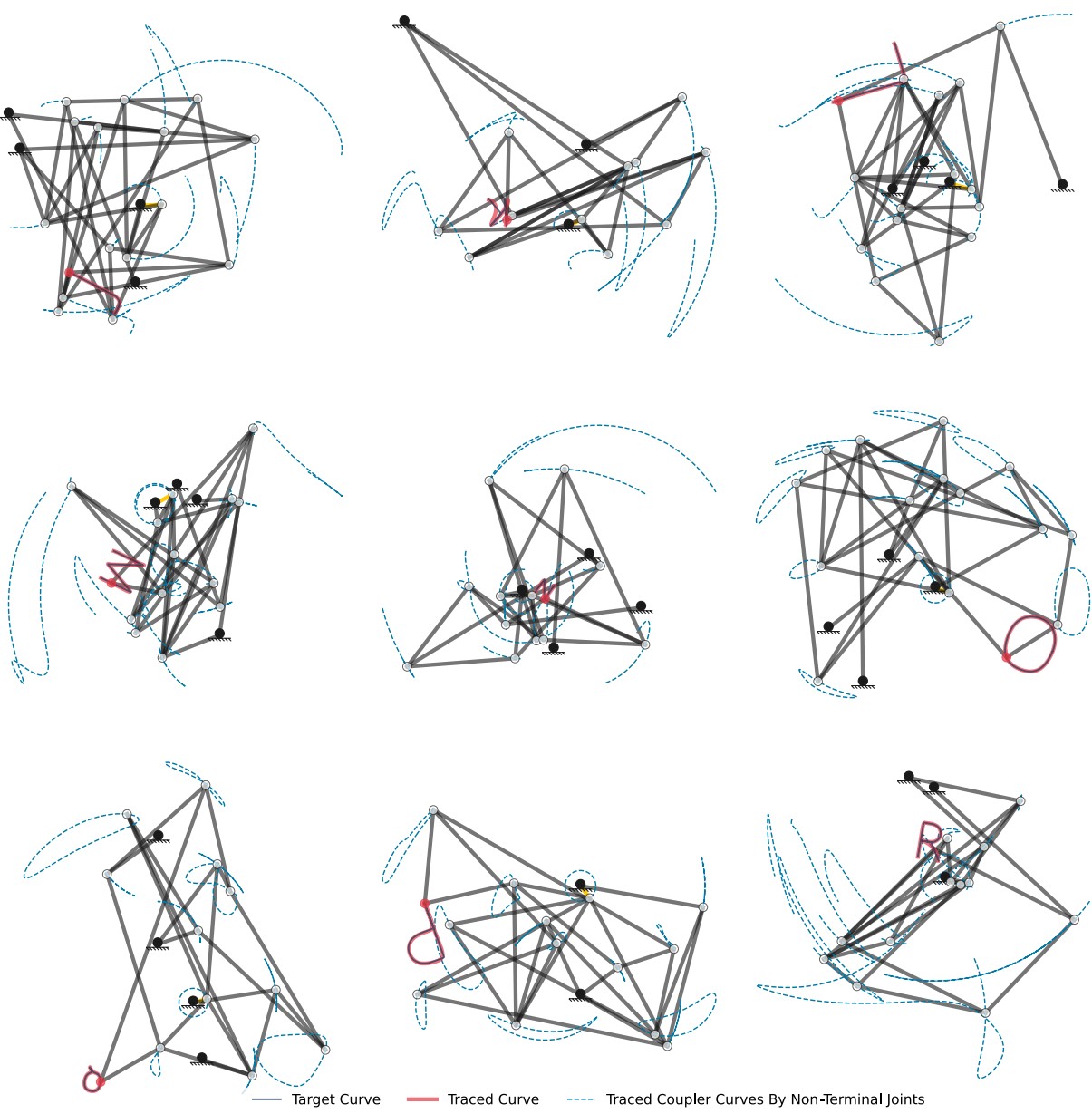

Figure 17: This figure shows the best results from LInK on the 9-18 alphabet test curves. As it is visible here, LInK has matched the target curves effectively without much challenge. Here, we visualize the mechanisms in 2D and also plot the paths traced by all the joints in each mechanism.

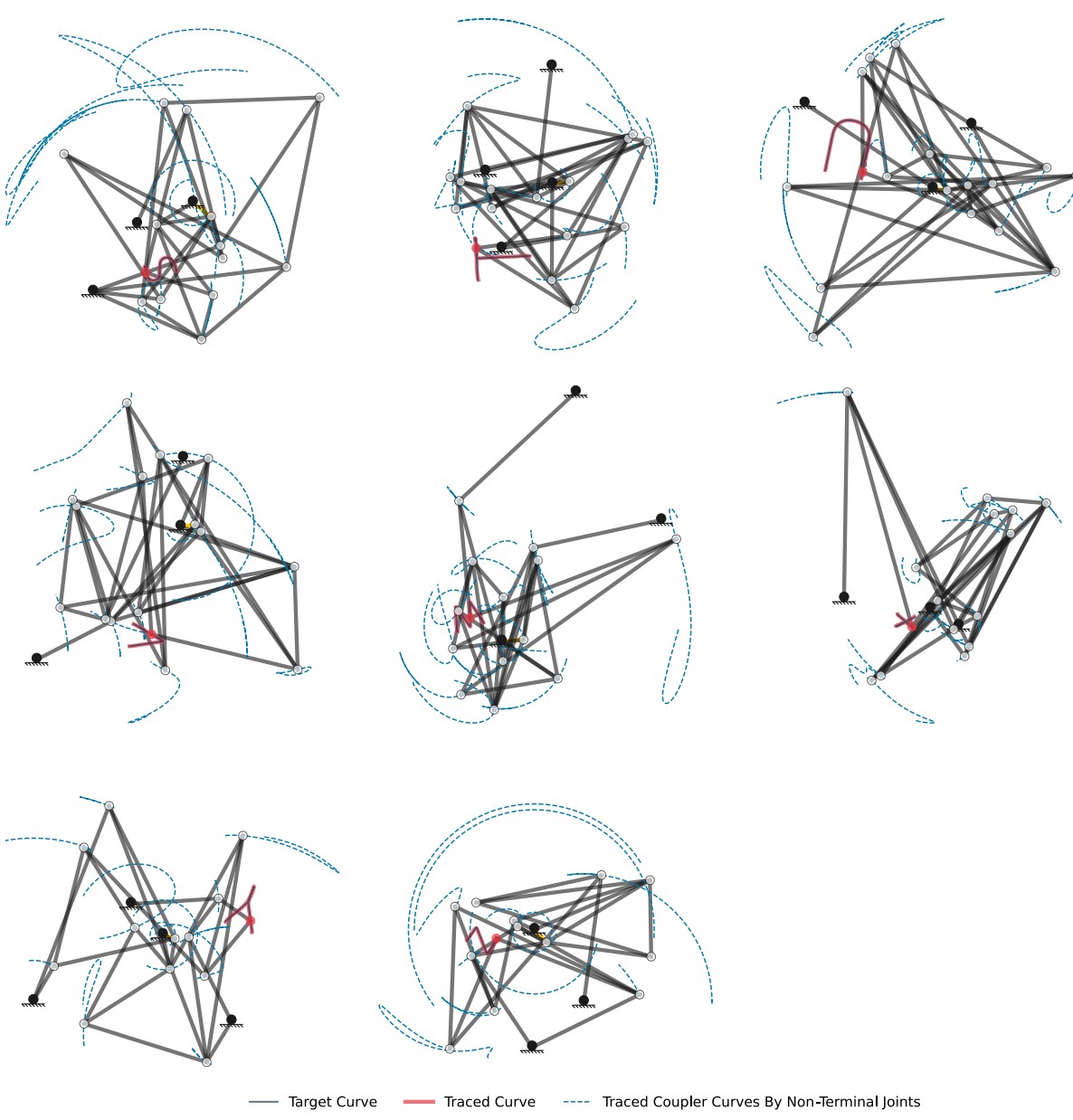

Figure 18: This figure shows the best results from LInK on the last 8 alphabet test curves. As it is visible here, LInK has matched the target curves effectively without much challenge. Here, we visualize the mechanisms in 2D and also plot the paths traced by all the joints in each mechanism.

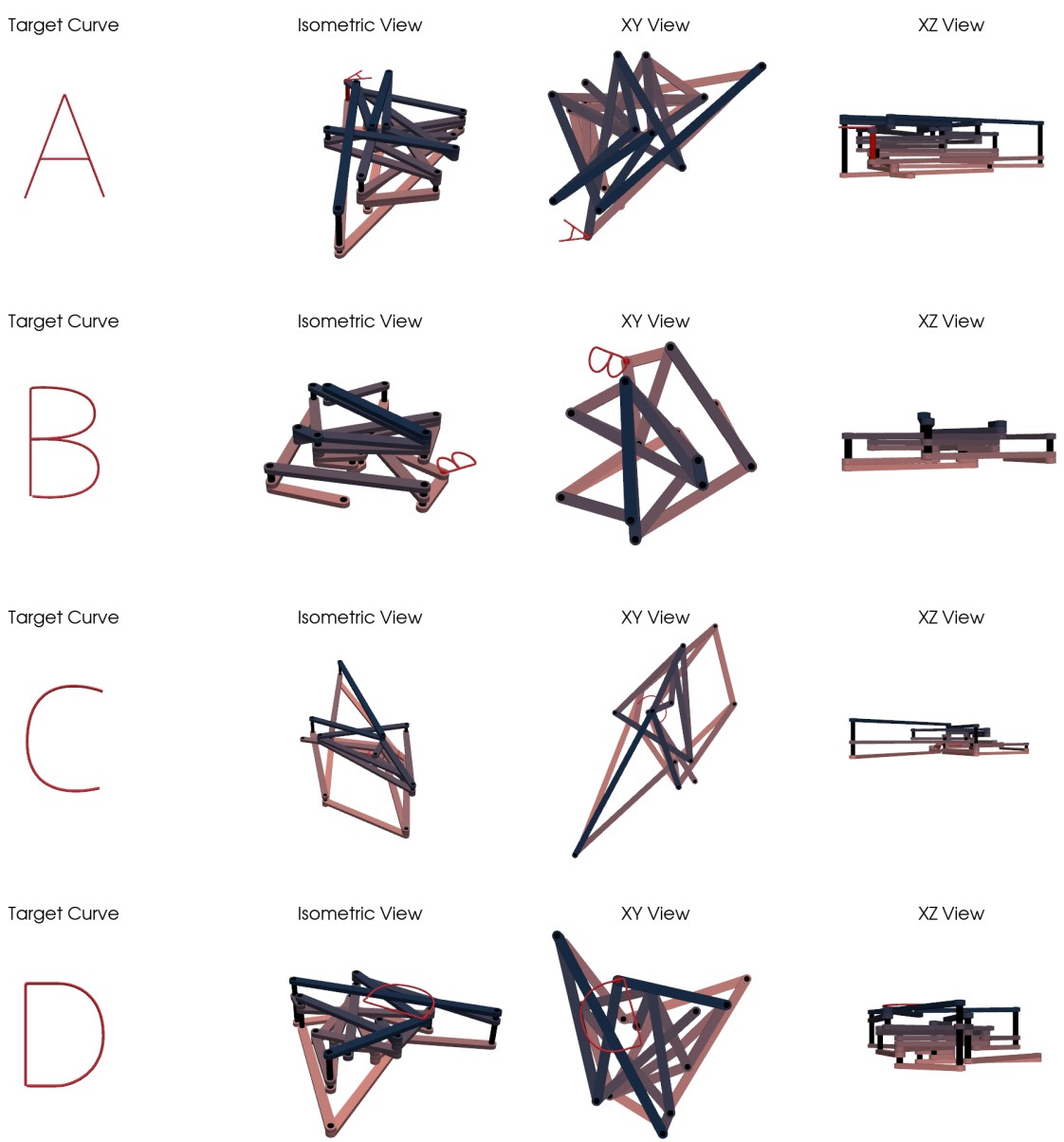

Figure 19: This figure shows the 3D optimal assemblies for mechanisms displayed in prior figures for the alphabet mechanisms (A-D).

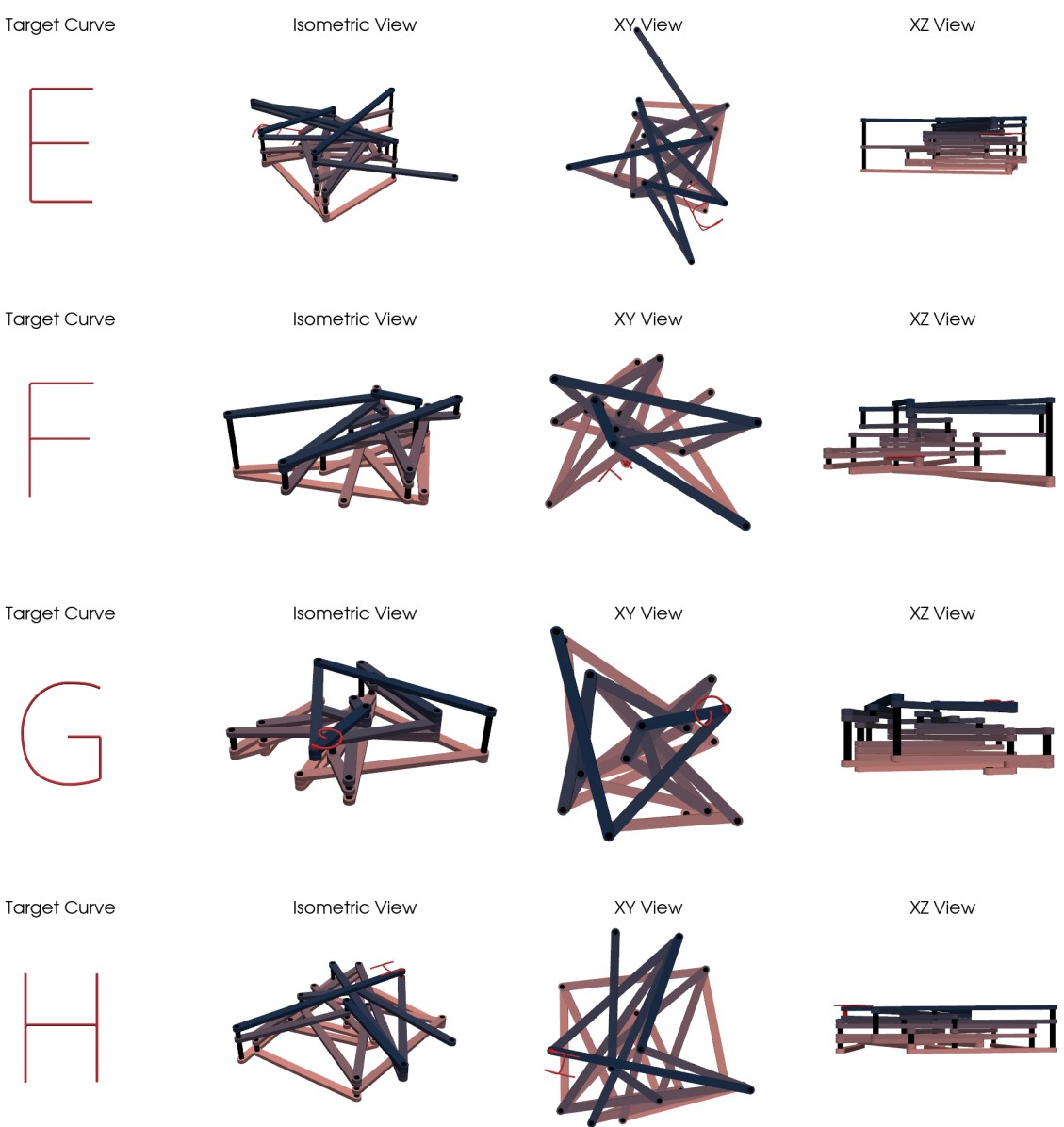

Figure 20: This figure shows the 3D optimal assemblies for mechanisms displayed in prior figures for the alphabet mechanisms (E-H).

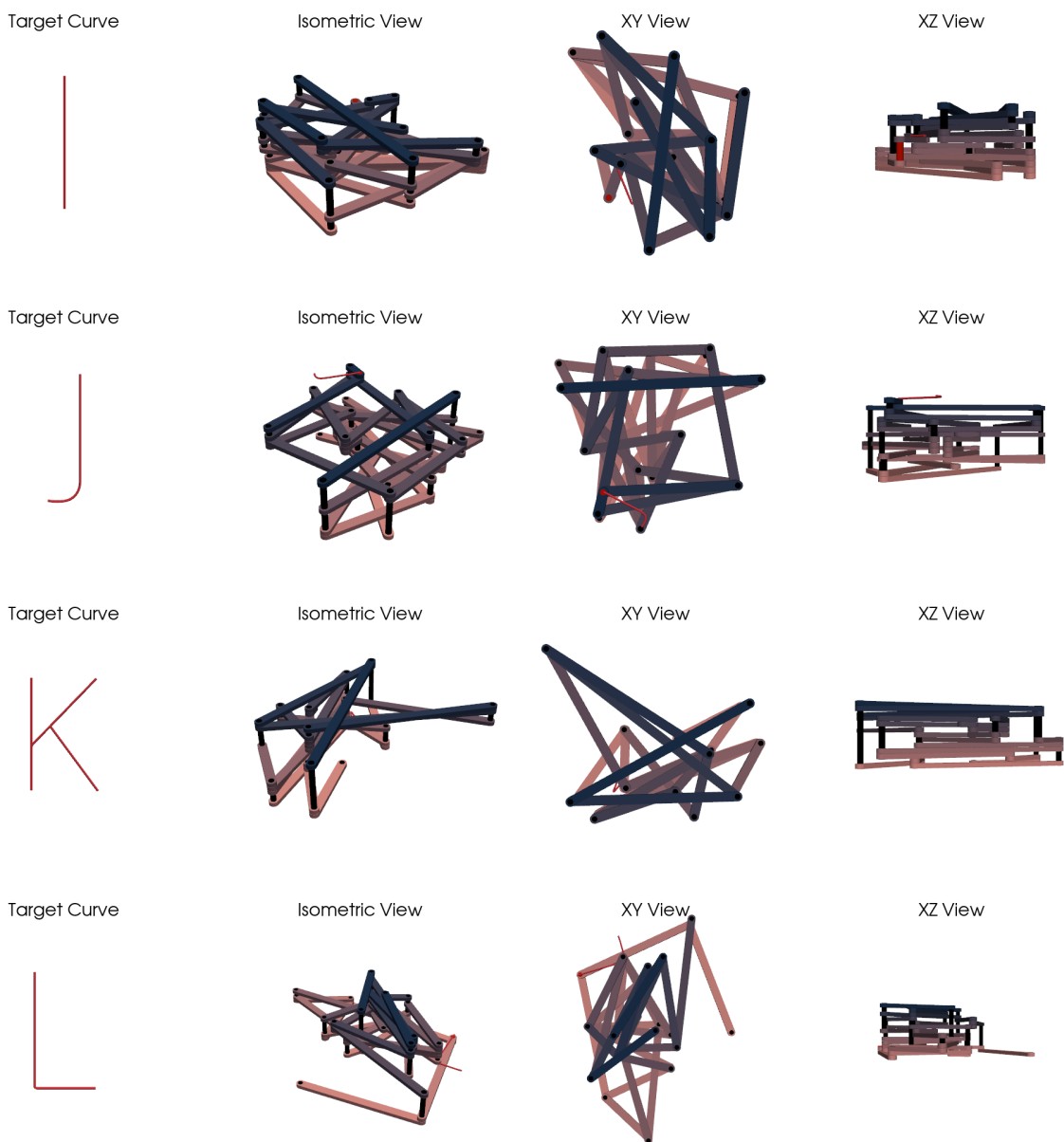

Figure 21: This figure shows the 3D optimal assemblies for mechanisms displayed in prior figures for the alphabet mechanisms (I-L).

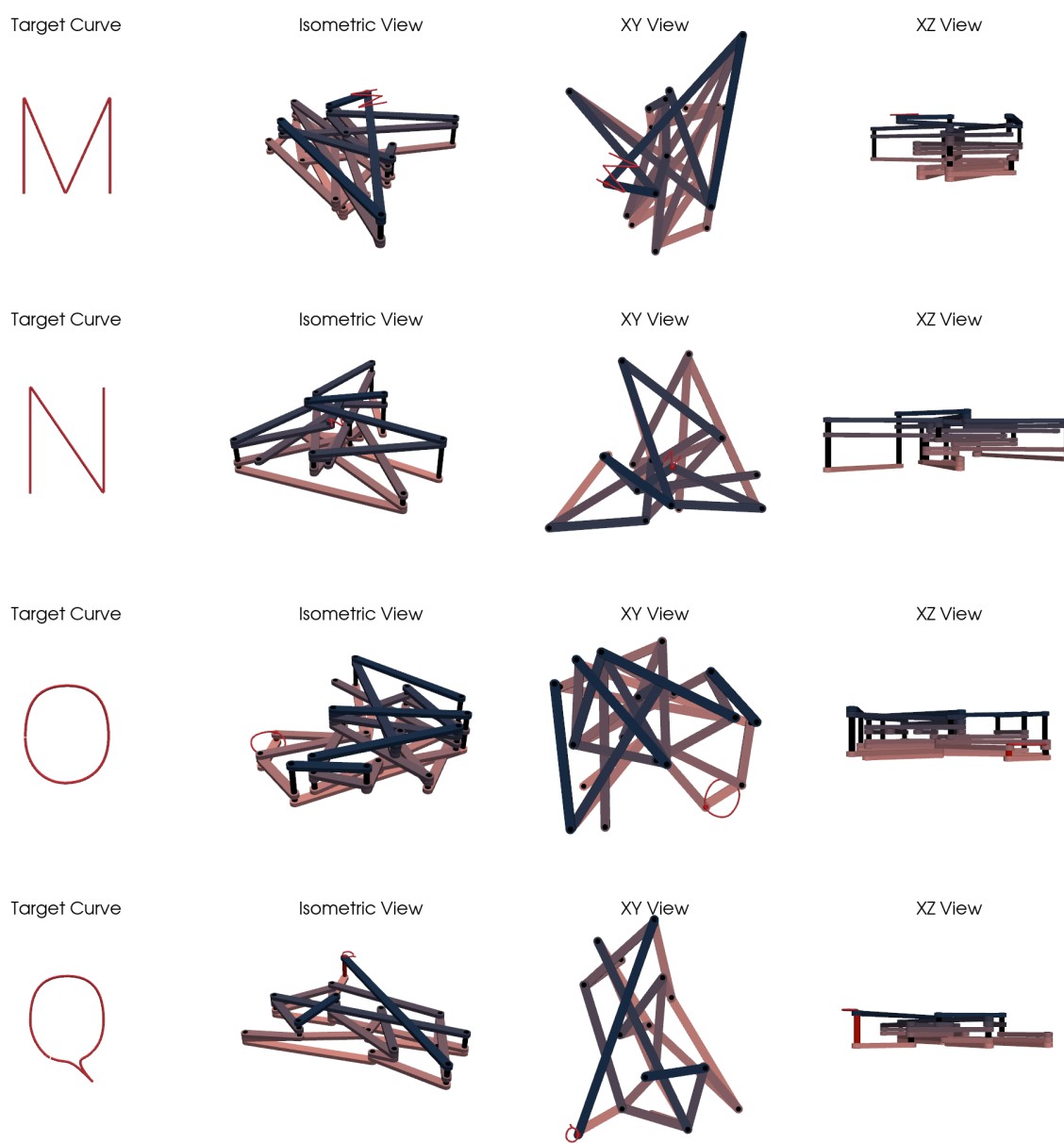

Figure 22: This figure shows the 3D optimal assemblies for mechanisms displayed in prior figures for the alphabet mechanisms (M-P).

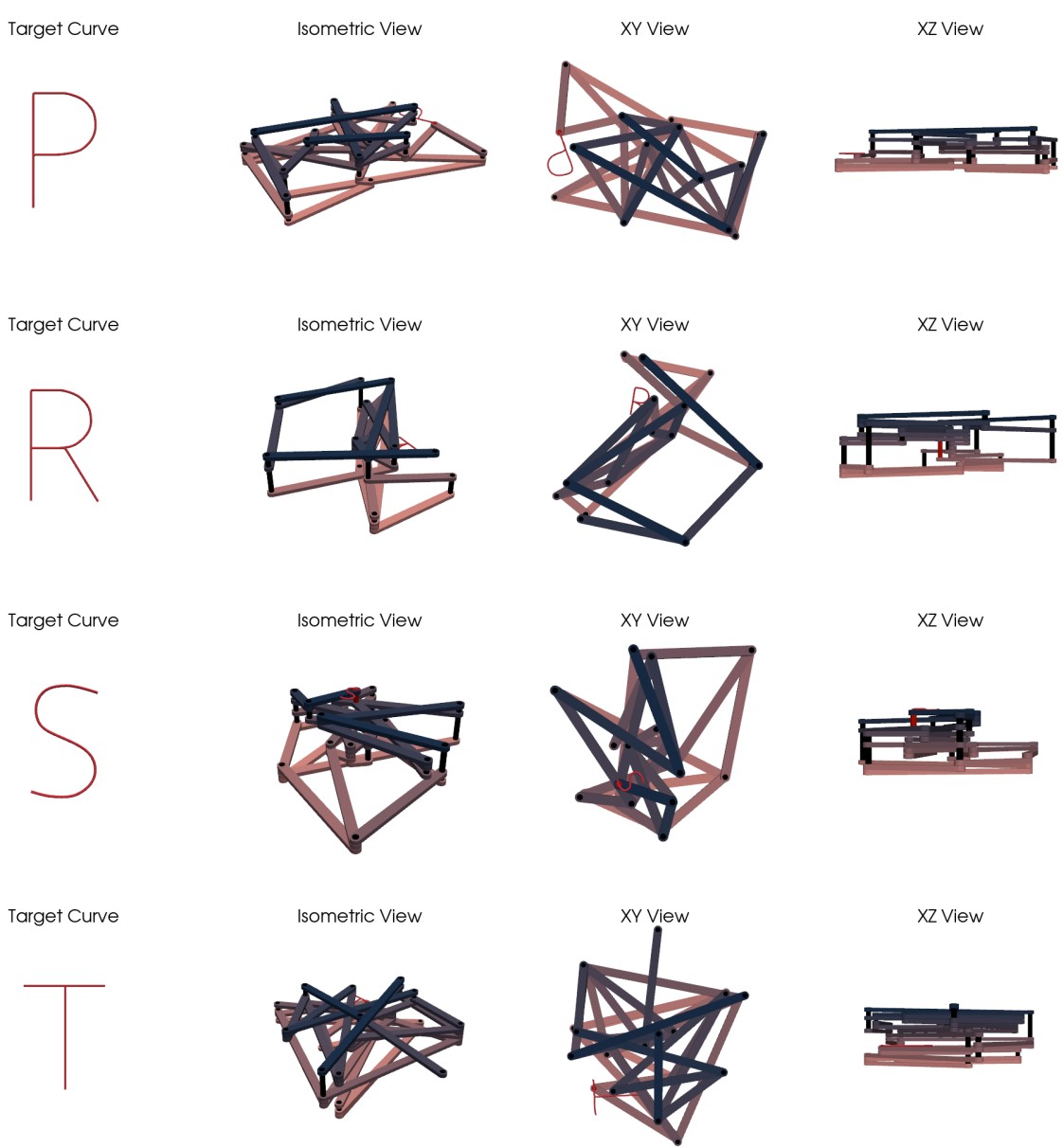

Figure 23: This figure shows the 3D optimal assemblies for mechanisms displayed in prior figures for the alphabet mechanisms (Q-T).

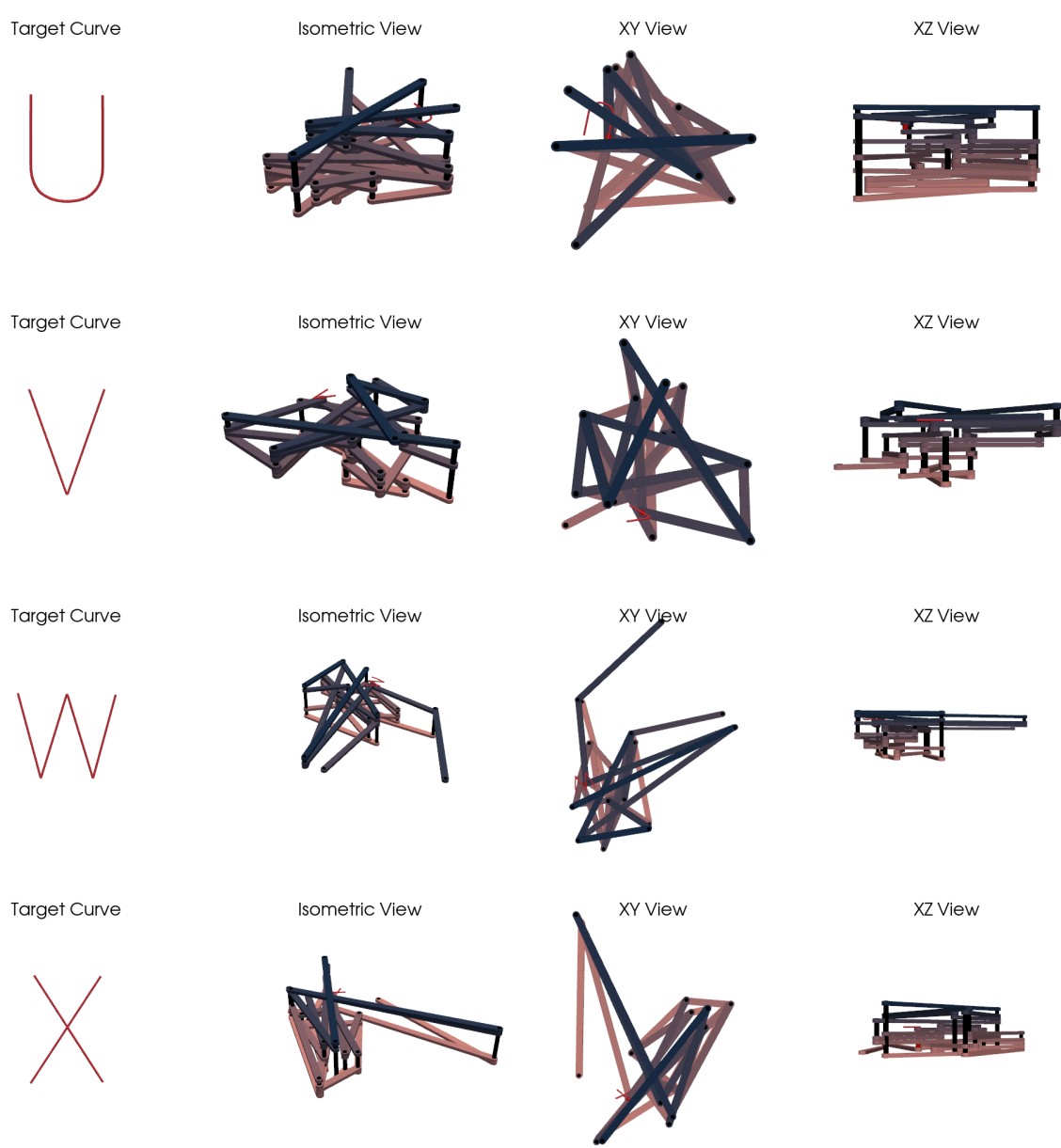

Figure 24: This figure shows the 3D optimal assemblies for mechanisms displayed in prior figures for the alphabet mechanisms (U-X).

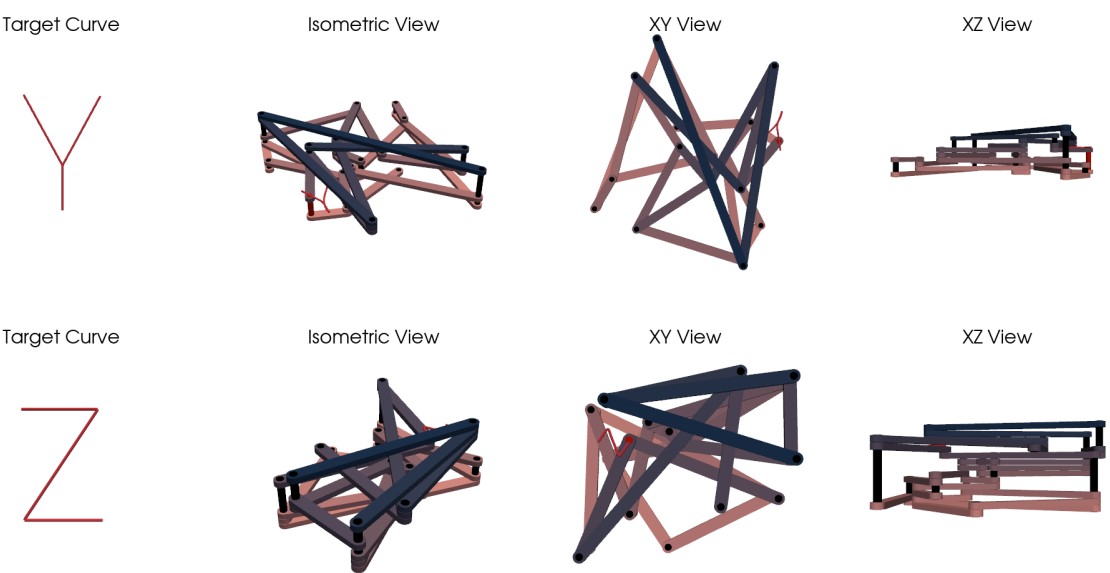

Figure 25: This figure shows the 3D optimal assemblies for mechanisms displayed in prior figures for the alphabet mechanisms (Y, Z).

