# OpenReview forum: "LInK: Learning Joint Representations of Design and Performance Spaces through Contrastive Learning for Mechanism Synthesis"
_TMLR — Accepted by TMLR_

### Review · Reviewer_g2oz · 2024-08-05

**Summary Of Contributions:**

The paper presents a new approach to kinematic synthesis and linkage design. The approach involves a contrastive learning framework for retrieval of candidate mechanisms and a subsequent optimization step for refining the mechanisms. The joint embeddings of mechanisms and curves learned by contrastive learning speeds up the search in the design space involving both continuous and combinatorial parameters. A novel graph network architecture with hop attention is designed to better capture the structure of mechanisms.
The proposed approach is shown to be faster and more accurate than existing methods.

**Audience:**

Yes

**Broader Impact Concerns:**

None.

**Claims And Evidence:**

Yes

**Requested Changes:**

See weaknesses.

**Strengths And Weaknesses:**

**Strengths**

I found the paper very interesting, both the problem and methodology.

The paper is very well-written, concise and clear.

An interesting graph network architecture that takes into account the structure of the kinematic problem.

Extensive experimentation supports the claimed improvements on accuracy and speed.


**Weaknesses**

Some methodology details could be better clarified. The advantage of contrastive learning can be better illustrated by comparing with a simpler baseline. Please see detailed comments and questions below.

The second contrastive loss, eq. 9, is a bit unclear. Why do the authors compute the contrasitive loss between full and partial curves, rather than between mechanisms and partial curves? How are the positive and negative pairs defined?

"For curve processing, we utilize two ResNet50 models (He et al., 2015): one handling full curves and another for partial curves."
How is the input data structured for the ResNet50 models?

Suggestion of ablation: An alternative to the contrastive learning approach is simply computing the similarity between the target curve and the curves in the dataset, and retrieve the closest curves and their corresponding mechanisms. How does the proposed contrastive learning approach compare to this baseline?

"For future work, the adaptability and robustness of LInK open new avenues for exploring its applicability across a broader spectrum of engineering challenges beyond linkage synthesis."
Can the authors provide some concrete examples of engineering problems that can benefit from the proposed methodology? The paper is focused on the kinematic synthesis problem, and giving such examples would be more precise in illustrating the potential and scope of the approach.

Please clarify what the number in LInK (8) through (16) means. Is it number of joints?

Eq. 2: is j = 1 to N correct? I think j != 1.

"Searching through such a massive dataset of this size by comparing curves and using conventional methods would be prohibitively expensive and time-consuming."
Did the authors compare with the methods in this fashion in experiments? Please provide more evidence to this claim.

There are a few typos that need to be fixed, e.g.:
- "In thtinuous nature and involves a non-linear solver whose solutions depend on bis example,"

- "We run our model for all three test cases and display the results visually in figures ??"

- "We also discussed the trade-offs of each metric with regards to comparing paths in Section 3.2.1. However, as far as comparison" (incomplete sentence)

---

> ### Author Response · Authors · 2024-08-31
> **Rebuttal Part 1**
>
> We sincerely thank the reviewer for their thorough and insightful evaluation of our paper. We are pleased that the reviewer found our work interesting, well-written, and clear, and that they recognized the novelty of our graph network architecture and the extensive experimentation supporting our claims.
> We appreciate the constructive feedback and the opportunity to clarify certain aspects of our methodology. The reviewer's comments have highlighted areas where we can provide additional context and explanation, which will undoubtedly strengthen the paper and enhance its value to the research community.
>
> ## On Writing Corrections
> We have specifically addressed the issues mentioned and a few more mistakes we found ourselves. We hope that this concern has been addressed adequately and we thank the reviewer for putting int the time to have a detailed readthrough of the paper.
>
> ## The Challenge With Partial Curves
> The second contrastive loss (Eq. 9) addresses the challenging problem of curve matching, which is particularly difficult when point correspondence is unclear. In curve matching, we must not only obtain the relative transformation of one curve to the other, but do so with point order in mind.
> For closed curves in our optimization stage, we use Procrustes analysis for scale and translation, but the critical step of rotation requires a brute-force search due to unknown correspondences. This involves testing 200 different relative rotations and measuring ordered distances for each, which is computationally intensive but feasible for closed curves.
> Partial and open curves present even greater challenges. We must identify which portion of a closed curve (traced by a mechanism) corresponds to the given partial curve. This isn't possible with brute-force methods since scale and translation are unknown, and we can't assume matching scale and overall shape. This complexity makes partial curve matching extremely difficult, highlighting the value of contrastive learning in effectively capturing this problem modality.
> We compute the contrastive loss between full and partial curves, rather than between mechanisms and partial curves, for several important reasons. Partial curves have a noisier signal overall. The positive and negative pairs for training are obtained by cutting out part of the full curves and deeming it positively associated with the full curve it comes from, and negatively associated with the rest of the full curves in the batch.
> The reason we don't directly incorporate partial curves with mechanisms lies in the fact that partial curves often contain common repeated patterns, namely arcs that are obtained from most shapes. When trying to associate mechanisms with partial curves, we found in our experiments that the learned representation's quality significantly decreases because of this. Most mechanisms will be associated with arcs while also being dissociated with curves of a similar geometry.
> This issue is less pronounced with curve-to-curve matching as more geometric information is available, such as curvature, which allows for cleaner matching between curves and partial curves obtained directly from said curves. This is not true with mechanisms, as the mechanism representation does not include any such information. Directly matching partial curves to mechanisms, although not impossible, does make the contrastive signal more noisy and hence reduces the representation capacity of the model. This is the primary reason behind our choice. We have modified 3.1.2 and 3.1.3 to better reflect these facts in our writing.

---

> ### Author Response · Authors · 2024-08-31
> **Rebuttal Part 2**
>
> ## Path Synthesis And The Use of CNNs
> For curve processing, we utilize two ResNet50 models: one handling full curves and another for partial curves. The input data structure for these ResNet50 models is designed to align with our preprocessing approach for path synthesis.
> We process all curves, both full and partial, to equidistant curves of the same length (200 points for contrastive learning). This standardization allows us to represent each curve as a sequence of 200 points, where each point has x and y coordinates. This standardization allows us to represent each curve as a 200x2 matrix, where each row corresponds to a point on the curve, and the two columns represent the x and y coordinates of that point. To adapt this for ResNet50 input, we reshape the 200x2 matrix into a 200x2x1 tensor, effectively creating a single-channel "image" where the x and y coordinates of each point are treated as the "height" and "width" of the image, respectively. This approach aligns well with CNN architecture capabilities.
>
> Our choice of CNN models, specifically ResNet50, is well-suited to this preprocessing approach. By using equidistant sampling and discarding timing information, we create a standardized representation that enables easy use of CNNs. This aligns with our focus on path synthesis, where we prioritize the geometric shape of the curve over specific timing or motion characteristics.
> There's a subtle nuance in processing curves as sequences that's worth noting: the lack of regularized permutations. Each curve's starting point is more or less arbitrary based on the mechanism that generated it. The same curve could appear in a slightly different mechanism with a different starting point, hence permutation of points. Some sequential models, particularly transformers with positional embeddings may struggle with this.
> While curve embeddings may not be so complex that sequential models couldn't handle this problem well, we have not tested sequential models in our studies. The main reason for using CNNs is that we are working on path synthesis, which allows us to discard timing and interpolate all curves in the dataset to a specific sequence length of equidistant points.
>
> To address this comment and provide additional clarity on our approach, we have made the following additions to our paper:
> 1. In section 3.1.2, we have added a brief description and reference to a new Appendix C when discussing timing invariance. This addition helps contextualize our approach within the broader scope of kinematic synthesis problems and clarify why we have fixed length curves.
> 2. We have included a new section (Appendix C) that elaborates on the different types of kinematic synthesis problems in planar linkages, including path synthesis, motion synthesis, and function generation. This section provides a comprehensive overview of these problem types, illustrated in a new figure (Figure 13).
>
> ## Table Caption On Numbers In Brackets
> We have updated Table 2's caption to clarify that the numbers are indeed joint counts. We have also adjusted the numbers to reflect direct comparison to prior methods with the same number of joint count limits. See also Figure 10 with more granualr analysis of solution quality with increased joint count.

---

> > ### Comment · Reviewer_g2oz · 2024-09-17
> >
> > "To adapt this for ResNet50 input, we reshape the 200x2 matrix into a 200x2x1 tensor, effectively creating a single-channel "image" where the x and y coordinates of each point are treated as the "height" and "width" of the image, respectively."
> >
> > Is the ResNet a 2D model? The ResNet structure might be an overkill for an input of 200x2x1, since there is little point to perform convolution along the dimension of size 2.

---

> > > ### Author Response · Authors · 2024-09-23
> > > **On CNN Architectures**
> > >
> > > Thank you for your insightful comment on the CNN architecture used in the paper. We do not focus on the architecture of our CNN model for curves since that is not the main contribution of our work. Furthermore, on the concern over ResNet-50 being too large, given the fact that we have 10M samples and we randomly transform them in each training step, not to mention that the partial curves are randomly selected and can be segments of any length the original curves, the model should not be unreasonably large for this size of data and augmentation. Most important thing to note is that we do not observe any significant over-fitting in our training. Given this and the fact that the focus of the paper is not on the CNN architecture of the curve model, we decided not to explore this aspect of the work in more depth.

---

> ### Author Response · Authors · 2024-08-31
> **Rebuttal Part 3**
>
> ## On Future Application Of The Approach
> Our framework for deep/generative optimization can be expanded, both in its current form and with modifications, to address several different engineering problems. Direct analogues to our problem exist in soft robotics and compliant mechanisms, where kinematic synthesis of a similar nature is investigated in non-rigid body components. Specifically in compliant mechanisms, there usually exists a mixed integer representation of the design space and a curve representation of the performance space with a differentiable solver with respect to continuous variables. As such, the proposed framework is directly applicable, and in our future research, we are actively working on this topic.
>
> Less direct analogues also exist in other optimization problems with high complexity, such as topology optimization. In these cases, there exist pseudo-continuous and differentiable solvers, yet for complex physics, the problem is not easily differentiable although local optimization is possible. If one were to capture the physics in a cross-modal contrastive space and find very good local candidates, this would make the problem feasible and significantly accelerate the process.
>
> To be more precise, the problem of topology optimization of structures for minimizing compliance (maximizing stiffness) is one that is easily done with gradient-based optimization. However, with complex physics, namely buckling, the formulation becomes highly complex and takes hours and days to solve. Yet the underlying physics remain to be a mixture of linear and/or plastic elasticity, which can be solved for random topologies fairly quickly, making a framework similar to ours very valuable. Demonstrating that such a framework can handle a problem that conventional optimization struggles with serves as great motivation for such a field.
>
> We appreciate this advice from the reviewer, and we have adjusted our concluding remarks to delve deeper into this matter and hopefully address the reviewer's concerns.
>
> ## On Manual Search In The Dataset
> We acknowledge that the value of the contrastive model-based retrieval may not be immediately apparent without quantifying the manual search speed. Two key points need clarification here.
>
> Firstly, there's a significant limitation of manual search when it comes to partial curves. We've discussed this in detail in our response to r4iV regarding the challenges of partial curve matching.
>
> Secondly, while we observe very good matching performance with our contrastive retrieval and might expect similar overall performance with a manual search (perhaps with a slight improvement given more precise similarity measures), the cost of manual search on 10M samples for each query is substantial. Moreover, there's a lack of any good mechanism for partial curve matching in manual search.
>
> Given these limitations, we don't consider manual search a truly viable baseline. However, we recognize the value of quantifying the speed of manual search to demonstrate the scale of improvement achieved by contrastive learning.
>
> To address this, we've made the following additions to our paper:
>
> 1. We've added the inference time for searching the dataset to Table 3, alongside the cost of searching using contrastive retrieval. This clearly illustrates the acceleration achieved by contrastive retrieval, in addition to its advantage in handling partial curves.
> 2. Table 3 now shows that GPU-based manual search takes 10,000 seconds for 10M samples, which is closer to MINLP and MICP, and much slower than even the RL-based method of GCP-HOLO. It's completely off from the 15 seconds of LInK. This performance gap is even worse if CPU is used for this purpose.
> 3. In contrast, contrastive search takes under 0.1 seconds, which is many orders of magnitude faster.
> 4. We've also added the time for Contrastive retrieval for more context and included a brief discussion on this in section 4.4.
>
> We believe these additions sufficiently address the reviewer's concerns by providing concrete evidence of the computational advantages of our contrastive learning approach over manual search methods.

---

> ### Comment · Reviewer_g2oz · 2024-09-17
>
> Thank the authors for their comprehensive response. I'd suggest including a link to a public repo containing the code in the final version.

---

> > ### Author Response · Authors · 2024-09-23
> > **Link to Code**
> >
> > We are planning on adding a link to GitHub upon acceptance. For now we avoid this given anonymous submission requirements.

---

### Review · Reviewer_r4iV · 2024-08-14

**Summary Of Contributions:**

The paper focuses on the path synthesis problem for planar linkage mechanisms which have discrete and continuous variables. This kinematic synthesis is a good representative of inverse problems encountered in engineering design. The key technical insight in the proposed solution is the combination of contrastive learning of design (mechanisms) /performance (curve) space with optimization methods.

The generative problem in engineering design has unique challenges:
- sparsity of data
- accurate capture of the physics needed for design
- high precision of prediction to guarantee satisfaction of the specification/requirements
- effective representation of design and performance space
- nonlinear and combinatorial nature of design

The design space is defined by the following:
- discrete: number of joints
- discrete: type of joints (fixed - restrict the linkage end to a stationary position in 2D space, revolute - allow free rotation while keeping the linkage endpoints fixed)
- discrete/topology: connection between joints using rigid linkages represented by adjacency matrix with n(n-1)/2 0/1 entries for n joints
- continuous: initial position of the joints

**Audience:**

Yes

**Broader Impact Concerns:**

There are no broader impact concerns.

**Claims And Evidence:**

Yes

**Requested Changes:**

Please fix the typos. For e.g., Figure 2 caption : "Demonstrates the path synthesis problem which involves a mixed combinatorial and conoth the combinatorial and continuous values."  + "In thtinuous nature", above eqn 10, "followingequation"

It is very distracting to have references that occur in a galaxy far far away. For e.g., Eqn 10 in Figure 2 reference is far away. Figure 7 on Page 2 shows up pages away.

The innovation in GHOP is not very clear. The idea appears to be we want to capture the fact that different nodes will take different steps for computation and usual GNNs won't be able to capture this. So, suggested solution is that let us use attention to learn to see which step is most relevant.

The timing aspect is not clear. It appears we are changing the problem when we force equidistant sampling of points on a curve. A strict interpretation of trying to draw a curve would be to also match the point sampling.

The idea of splitting batch optimization in two steps - first for 10 steps and then for a selected few for 150 steps, seems a bit hacky. Any intuitive justification for why it is expected that one would be able to find "hopelessly flawed and so discardable designs" within 10 steps.

A related question is that what is difficult about partial curve tracing. It is called out as "unique contribution" but it is not clear what is particularly challenging with it.

Why not model curve as a sequence rather than use Resnet50 ?

**Strengths And Weaknesses:**

+ The paper includes creation of over 10 million mechanisms - this is a vast dataset that would aid evaluation of ML techniques being developed for inverse problems in engineering design. Even though it is a much simpler kind of engineering problem, this is a good start.

+ The improvement in accuracy and speed is very significant when compared against SOTA.

+ Linkage mechanism appears to be a simple kinematics problem but has applications such as robotics, auto engines, printing, biomechanics, computer graphics, etc.

+ The use of multimodal transformation-invariant contrastive learning of joint embeddings is very interesting.

+ Using embedding based search for warm-start of optimization method is a good practically useful trick.

---

> ### Author Response · Authors · 2024-08-31
> **Rebuttal Part 1**
>
> We are deeply grateful for the thorough and thoughtful review provided for our paper. The reviewer's careful examination of our work and recognition of its key contributions is much appreciated. We value the acknowledgment of our dataset's significance and the noted improvements in performance over existing methods. The reviewer's feedback, is very helpful and we will address each point in detail below, hoping that these insights will help further improve our work and address the concerns brought up.
>
> ## On Writting Corrections
> We have specifically addressed the issues mentioned and a few more mistakes we found ourselves. We hope that this concern has been addressed adequetly and we thank the reviewer for putting int the time to have a detailed readthrough of the paper.
>
> ## Figure and Equation Refrences In The Text
> We origianlly, had the mentioned figures separately included in the closer sections and combined the figures for better visual appeal and brevity in the length of the paper. But we realize that this may have come at the cost of readablity. Given this we have gone back to the original separate figures. We hope this has adequetly addressed the concerns of the reviewer on this matter.
>
> ## GHop And Its Intuition
> The reviewer's intuition about GHop is correct. Traditional graph neural networks apply message passing uniformly across all layers, with the receptive field of output features for each node determined solely by the number of layers. However, as illustrated in Figure 5, our mechanism solver only processes nodes until they are solved, after which no further analysis is required.
> GHop addresses this unique aspect by allowing the deep learning model to adapt the receptive field for each node individually through an attention mechanism. This approach enables the model to incorporate the specific solution characteristics of linkage mechanisms in an unsupervised manner. Our results demonstrate that GHop outperforms baseline GNN models, indicating an improved representation of linkages.
> To clarify this concept for readers, we have revised the second paragraph in section 3.1.1, providing a more detailed introduction to the model and its advantages in the context of linkage mechanism synthesis.
>
> ## The Timing Information In Path Synthesis vs. Motion Synthesis
> We appreciate the reviewer's insightful observation regarding the timing aspect of our approach. The reviewer is correct in noting that our use of equidistant sampling of points on a curve represents a specific interpretation of the path synthesis problem. Our focus in this work is primarily on the geometric aspects of path generation, where the goal is to create a mechanism that can trace a desired path, regardless of the specific timing or speed of the motion.
>
> We acknowledge that this method does not capture the full complexity of motion synthesis, where timing and speed are integral components. However, for the purposes of our current study and the development of our framework, we believe this simplification provides a robust foundation for addressing the core challenges of path synthesis. Our decision to use equidistant sampling is a deliberate choice to standardize the representation of curves across different mechanisms. This approach allows for more consistent comparisons and evaluations of the generated paths, irrespective of the specific timing characteristics of individual mechanisms. While we acknowledge that this may deviate from a strict interpretation of curve drawing that includes specific positions and sometimes timing information, it aligns with the core objectives of path synthesis in linkage mechanism design. We believe this approach strikes a balance between practical applicability and the need for standardized evaluation in our machine learning framework. However, we recognize the validity of the reviewer's point, and in future work, we could explore extensions of our method that incorporate timing information for problems where it is critical, such as motion synthesis.
>
> To address this comment and provide additional clarity on our approach, we have made the following additions to our paper:
> 1. In section 3.1.2, we have added a brief description and reference to a new Appendix C when discussing timing invariance. This addition helps contextualize our approach within the broader scope of kinematic synthesis problems.
> 2. We have included a new section (Appendix C) that elaborates on the different types of kinematic synthesis problems in planar linkages, including path synthesis, motion synthesis, and function generation. This section provides a comprehensive overview of these problem types, illustrated in a new figure (Figure 13).
>
> These additions aim to clarify the scope of our work and provide readers with a broader understanding of the various problems in linkage mechanism synthesis. We believe this context will help readers better appreciate the specific focus of our study.

---

> ### Author Response · Authors · 2024-08-31
> **Rebuttal Part 2**
>
> ## On Optimization Steps
> We acknowledge that there is no direct justification for the choice of number of steps. This choice is primarily an engineering decision and an optimization algorithm hyper-parameter. The intuition behind it stems from the nature of our problem: we're dealing with a highly non-linear and non-convex space containing numerous local minima.
> Our approach leverages contrastive learning, which allows us to quickly retrieve candidates from a massive dataset of 10M that are more likely to have favorable local minima suited to the target path. This retrieval step is what makes our method powerful, as it efficiently narrows down the search space.
> Given this initial selection, combined with the fact that BFGS typically converges rapidly towards local minima, we posit that a few steps of BFGS should provide substantial information about a design's potential for good precision in its local minimum. Specifically, 10 steps were chosen as sufficient to identify more promising candidates and not to eliminate "hopelessly flawed and so discardable designs" since the remaining could converge albeit at a slower rate to a similarly good solution.
> However, it's important to note that this is purely an engineering choice, empirically tested in our experiments. We don't claim it yields the optimal outcome for optimization, but rather a pragmatic balance between accuracy and computational efficiency.
> The number of initially retrieved candidates and the iteration counts (10 initial, 150 for selected designs) are also engineering choices, largely dictated by the desired solution time. In principle, one could run the optimization on a larger retrieval set with more iterations, potentially achieving even higher accuracy at the cost of increased computation time.
> Our method demonstrates that it's capable of producing highly accurate results with the chosen parameters. The trade-off between accuracy and speed can be adjusted by modifying these parameters based on specific requirements.
> In essence, while this two-step approach might seem somewhat arbitrary, it's a practical solution tailored to the challenges of our problem space, consistently delivering accurate results as evidenced by our experimental outcomes. We have clarified this by modifying section 3.3 and adjusting the language in that section.
>
> ## The Challenge With Partial Curves
> The reviewer raises an interesting point. Curve matching in general is a rather difficult problem, especially when point correspondence is not clear. That is to say, we do not know which point in one curve corresponds to which point in the other. As such, curve matching is sometimes done with Chamfer distance and brute-force search or Iterative Closest Point (ICP). However, this works fine when order of points is not important. In curve matching, one must not only obtain the relative transformation of one curve to the other, but this must be done with order in mind.
> For example, for closed curves in our optimization stage, we find the relative scale and translation fairly easily with the first two steps of Procrustes analysis. However, the third and most important step of Procrustes analysis requires known correspondence, which we do not have for any two given curves. Therefore, we find the rotation with a brute-force search. This means we try 200 different relative rotations and measure ordered distance for each of them. The computation of ordered distance involves testing all possible ordered correspondences between two curves, which is equal to the number of points in the curves (given we do not have combinantion of points and the order of points is dictated). This is all possible for closed curves.
> Once we have partial and open curves, we face the problem of finding which portion of a closed curve traced by a mechanism is going to correspond to the partial curve at hand. This is not even possible with brute-force since the scale and translation are not known, as the first two steps of Procrustes analysis assume matching scale and overall shape. However, we cannot even know the part of the closed curve we want to achieve this. This is what makes partial curve matching so difficult, and the fact that contrastive learning is able to capture this modality of the problem effectively is of great value. Without this, finding matches among closed curves conventionally would prove either extremely burdensome computationally (brute-force at every level) and rather difficult to solve directly through a mixture of optimization techniques such as ICP (notoriously sensitive to initialization) and brute-force search.
> We agree that the importance of this contribution may not be appreciated by readers immediately. As such, we have added a brief paragraph in section 3.1.2 to clearly convey the importance of partial curves. We thank the reviewer for pointing this out and helping to improve the quality of the paper.

---

> ### Author Response · Authors · 2024-08-31
> **Rebuttal Part 3**
>
> ## The Use of CNNs For Curve Embeddings
> We appreciate the reviewer's suggestion about modeling curves as sequences. Indeed, curves can be processed as sequences since they contain order, and using sequential models for curve embeddings is a valid approach to the problem.
> However, our choice of CNN models, specifically ResNet50, is well-suited to our preprocessing approach. We process all curves to equidistant curves of the same length (200 points for contrastive learning), which aligns well with CNN architecture capabilities.
> There's a subtle nuance in processing curves as sequences that's worth noting: the lack of regularized permutations. Each curve's starting point is more or less arbitrary based on the mechanism that generated it. The same curve could appear in a slightly different mechanism with a different starting point, hence permutation of points. This characteristic might pose challenges for some modern sequential models, particularly transformers with positional embeddings.
> That being said, curve embeddings may not be so complex that sequential models couldn't handle this problem well. However, we have not tested sequential models in our studies.
> The main reason for using CNNs is that we are working on path synthesis. As discussed earlier, this allows us to discard timing and interpolate all curves in the dataset to a specific sequence length of equidistant points, which enables easy use of CNNs.
> We hope this explanation satisfies the reviewer's query about our choice of model architecture.

---

> > ### Comment · Reviewer_r4iV · 2024-09-01
> > **Thank you!**
> >
> > The rebuttal has addressed the major concerns of the reviewer.

---

### Review · Reviewer_xBAh · 2024-08-17

**Summary Of Contributions:**

This manuscript presents a novel framework dubbed LInK, that integrates contrastive deep learning with traditional optimization techniques to tackle complex inverse design problems in engineering, with a focus on the path synthesis problem for planar linkage mechanisms.

The LInK framework introduces a bimodal, contrastive learning approach to create joint embeddings between mechanism designs and paths, allowing for efficient retrieval of candidate mechanisms given a path. The resulting candidates are then used to further refine menchanism designs using traditional optimization methods, with the resulting multi-step process leading to significantly improved performance in speed and accuracy compared to existing methods. The authors also propose a new, more challenging benchmark named LINK-ABC, which involves synthesizing linkages that trace the trajectories of English capital letters, with the aim of pushing the current state-of-the-art to a higher standard.

**Audience:**

Yes

**Broader Impact Concerns:**

There are no broader impact concerns as far as I am aware.

**Claims And Evidence:**

Yes

**Requested Changes:**

Requested Changes
- If my understanding is correct, the subscript i Equation 4 should be corrected to \nu, as it seems to refer to the current current joint. This is a minor but necessary correction for clarity and accuracy.
- On first paragraph on page 15, the final sentence seems to be cut off, this needs to be corrected.
- The final paragraph on page 16 contains a faulty reference to a figure, I believe it should be a reference to figure 7.
- One set of results the authors might want to include is a more 'fair' comparison to the state-of-the-art methods by limiting the LInK framework to subsets of mechanisms with 7 and 11 joints, instead of the selected 8 and 16. This is because, as the authors themselves report, the competing methods MINLP and MICP can design mechanisms up to 7 joints, and GCP-HOLO up to 11. While providing a more fair comparison in the constraned scenarios, these results will in no way diminish the significantly better performance achieved by LInK in the unconstrained case, because being efficient enough that it can design mechanisms with a much larger number of joints is part of the strengths of LInK. This change is not neccesary for acceptance, but it would provide a better understanding for the performace of LInK's optimization procedure, by decoupling it from computational efficiency.
- Another result that can be included in the supplement is how the performance of LInK changes when manufacturability is excluded. This could provide a better understanding of how the error of the other models is affected, since they do not do any filtering based on manufacturability. This change is not necessary for acceptance, but I believe it would make a stronger case for the LInK framework regardless of the result.

**Strengths And Weaknesses:**

Strengths:
- The paper is clearly written, well structured and very easy to read. The explanations are both straightforward and detailed, making the complex concepts accessible to a broader audience, including those who might not be familiar with the field of kinematic synthesis, such as myself. The background section particularly useful in this regard, providing a good overview of the problem, detailing both traditional and modern approaches. This context is valuable for readers new to the topic and helps in understanding the significance of the proposed contributions. The methods such as: the contrastive learning framework, the novel graph neural network architecture, and the optimization process, are also explained well and in great detail, giving the reader a clear understanding of all the working parts of the proposed method.
- The performance evaluations are detailed and the results are very impressive. The paper compares LInK with state-of-the-art methods on the standard benchmark, providing very impressive improvements both in terms of accuracy and speed. On top of that, the paper also provides two new more extensive benchmarks, one taken from the same distribution as the training set, and another more challenging benchmark consisting of the letters of the English alphabet, which are significantly different to anything found in the training data. The LInK framework performs very well on both these benchmarks, which sets a high standard for future work in this task.
- The novel GHop neural network architecture is also very intriguing, with the included ablation analysis (which is particularly welcome) and showing that it significantly outperforms other standard graph neural network architectures. This definitely justifies the choice of the authors for this new architecture, but it would also be interesting to hear a more detailed discussion (beyond the brief) of what other problems the authors think this architecture could be particularly useful for.

Weaknesses:
Overall, the paper is very compentently written and does not have any major weaknesses. The one notable aspect that could be considered a weakness is the lack of an inherently invariant representation for curves. While the authors addresses invariances through rescaling and rotational data augmentation on the target curves, having an inherently invariant representation might further strengthen the robustness of the method.

Aside from this, there are some minor errors, typos and omissions that I will address in the Requested Changes section.

---

> ### Author Response · Authors · 2024-08-31
> **Rebuttal**
>
> We extend our sincere thanks to reviewer xBAh for their comprehensive and insightful evaluation of our manuscript. We appreciate the recognition of our paper's clarity and our framework's performance. The reviewer's constructive feedback, including suggestions for additional analyses and minor corrections, is invaluable. We will address each point in detail below, incorporating these insights to further strengthen our work.
>
> ## On Inherently Invariant Representation
> We appreciate the reviewer's insight regarding an inherently invariant representation. While such a representation could indeed improve the model by allowing it to focus more on contrastive aspects rather than learning invariance, truly invariant representations of curves present significant challenges.
> Scale and translation invariance are relatively straightforward to achieve using the first two steps of Procrustes analysis. However, rotation invariance is more complex. One approach is to align the principal axes of curves, but this method has inherent ambiguities. For instance, aligning a principal axis with the x-axis can be achieved by two rotations 180 degrees apart.
> More critically, the principal axis (defined as the direction from the center to the furthest point) can be highly sensitive to small perturbations in the curves. In many cases, multiple parts of a curve might be equally distant from the center, or have very small differences, leading to ambiguity in determining the principal axis. This issue is further complicated when dealing with partial or open curves.
> Given these challenges, it's difficult to achieve a fully unambiguous invariant representation. Learning invariance through our current approach proves more robust compared to potentially ambiguous normalization methods. Our method does incorporate normalization for scale and translation, as these transformations are unambiguous.
> We believe this approach strikes a balance between leveraging known invariances and allowing the model to learn more complex invariances from the data. However, we acknowledge that further research into robust invariant representations could be valuable for future work in this area.
>
> ## On Writing Corrections
> Thank you for your careful review and for pointing out several important corrections:
>
> 1. Regarding Equation 4: We appreciate you bringing this to our attention. The equation has been rewritten and corrected. This error occurred during a revision process where we changed some subscripts, leading to an incorrect edit. We're grateful for your keen eye in catching this mistake.
> 2. Incomplete sentence and faulty figure reference: These issues have been addressed in our revised manuscript. The incomplete sentence has been completed, and the figure reference has been corrected to accurately point to the intended figure.
> 3. Overall proofreading: Prompted by your thorough review, we have conducted a comprehensive read-through of the entire paper. We have addressed all grammar and spelling issues to the best of our ability, including those you and other reviewers have pointed out.
>
> We sincerely appreciate your attentive reading of our manuscript and for highlighting these errors. Your feedback has been invaluable in improving the clarity and accuracy of our paper.
>
> ## On Joint Count Experiments
> We acknowledge the limitation in our analysis. We have updated Table 2 with the requested changes and Figure 8 has been updated to include more granular results of joint count and its effect on solution quality. We have also adjusted the language in section 4 to describe the results better and the methodology of the experiment we use to obtain results for Table 2.
>
> ## On Manufacturability Effect
> we must say that most mechanisms that are obtained are manufacturable and in fact, in our experiments, the best performing mechanisms in Table 2 are indeed manufacturable, to begin with, and only a small number of the obtained mechanisms in the other test dataset experiments tend to be eliminated because of manufacturability. As such the performance is not significantly different, and as such it realistically does provide much insight for the readers. The contribution of the paper is formulating the manufacturability problem and introducing the filtering through MILP. Given this insight if the reviewer still deems it an important experiment we can add these results to the appendix in a later revision.

---

### Author Response · Authors · 2024-08-31
**Overall Response To Reviewers**

We would like to extend our sincere gratitude to all the reviewers for dedicating their valuable time and expertise to provide us with insightful feedback. Your thorough reviews have significantly contributed to improving our work. We truly appreciate your effort and the comprehensive nature of your comments.
Below is a summary of the changes made and the main points addressed in our response:

## Important Changes To Results And Inference Speed
Since the initial submission of our manuscript, we have made significant strides in enhancing the computational efficiency of our method. By leveraging the JAX library in Python, we have substantially improved the speed of both our simulations and the BFGS optimization process. This optimization has resulted in a sixfold increase in inference speed compared to what was originally reported in our paper.
We have updated our supplementary materials to include the revised code implementing these improvements. With this enhancement, we believe our method now achieves real-time path synthesis, marking a significant advancement in the field.
These updates have been incorporated into the revised manuscript, providing a more current representation of our method's capabilities. We believe this improvement further underscores the practical applicability and efficiency of our approach.
With these improved speeds we are able to run twice as many iterations in our BFGS line search while retaining the 6 fold speed up, significantly improving our performance as well speed. These results have also been reflected in our new quantitative results in Tables 1, 2 and 3 and Figure 8.
We also agree with the point brought up regarding performance comparisons in Table 2 with lower joint count runs of LInK. As such we have adjusted them to 7 and 11 joints which MINLP/MICP and GCP-HOLO use respectively.
Finally, inference time for only retrieval using manually searching the dataset and contrastive retrieval have been added to Table 3 to better highlight the value of contrastive retrieval in this study with these results demonstrating without doubt how important the contrastive learning retrieval is with many orders of magnitude faster retrieval.

## Regarding Spelling/Grammar And Proofing
We have addressed the issues brough up by all reviewers and a few more mistakes we found in the writting in our read through and have fixed those issues to the best of our abilities. We hope that this has improved the writting quality of the paper and addressed the concerns of reviwers in this regard.

## Highligthed Changes
We have provided the latexdiff for the paper revision in the supplamentary materials for reviewers to see the adjustements more easily. We hope that this can expedite any requests for further revisions and chnages before the decision deadline of Sep 14.

---

### Decision · Action_Editor_P2JJ · 2024-09-24

**Recommendation:** Accept as is

**Comment:**

The authors have properly addressed the reviewers' comments, after which all reviewers recommend acceptance.  The AE agrees.

**Audience:**

Yes, the sub-community on AI for design will be interested in this work.

**Claims And Evidence:**

Yes, all reviewers agreed that the submission presents an interesting and useful mechanism for linkage design, as well as a benchmark.